# Comparative analysis of squamate brains unveils multi-level variation in cerebellar architecture associated with locomotor specialization

Simone Macrì [1], Yoland Savriama [1], Imran Khan[1] & Nicolas Di-Poï [1]*

Ecomorphological studies evaluating the impact of environmental and biological factors on the brain have so far focused on morphology or size measurements, and the ecological relevance of potential multi-level variations in brain architecture remains unclear in vertebrates. Here, we exploit the extraordinary ecomorphological diversity of squamates to assess brain phenotypic diversification with respect to locomotor specialization, by integrating single-cell distribution and transcriptomic data along with geometric morphometric, phylogenetic, and volumetric analysis of high-definition 3D models. We reveal significant changes in cerebellar shape and size as well as alternative spatial layouts of cortical neurons and dynamic gene expression that all correlate with locomotor behaviours. These findings show that locomotor mode is a strong predictor of cerebellar structure and pattern, suggesting that major behavioural transitions in squamates are evolutionarily correlated with mosaic brain changes. Furthermore, our study amplifies the concept of 'cerebrotype', initially proposed for vertebrate brain proportions, towards additional shape characters.

[1] Program in Developmental Biology, Institute of Biotechnology, University of Helsinki, 00014 Helsinki, Finland. *email: nicolas.di-poi@helsinki.fi

Understanding the processes underlying the origin and diversification of morphology is a central goal in evolutionary biology and requires the integration of form, function and ecology. Particularly, there is compelling evidence across diverse vertebrate species that behavioural and ecological factors such as diet, habitat, locomotion, cognitive abilities and lifespan play an important role in driving brain evolution[1]. Previous studies evaluating the impact of environmental and biological factors on morphological brain characteristics have so far largely focused on neuroanatomical descriptions or volumetric, linear, and/or stereological measurements of whole-brains, major brain subdivisions, or brain endocasts. However, very few investigations have integrated multiple approaches to study brain evolution at various levels of biological organisation, ranging from molecules to cells and organ shape, within specific clades of vertebrates.

The cerebellum—major hindbrain feature—plays a critical role in sensory-motor control and higher cognitive functions such as memory and language[1]. Furthermore, this brain subdivision shows an extensive diversity in terms of morphology and neuroanatomy across vertebrates[1–3]. This suggests that cerebellar organisation is likely tightly linked to functional demands associated with ecological behaviours. Comparisons of vertebrate cerebella using complementary approaches could thus offer an excellent framework to address the complex evolutionary relationships between organ specialization and behavioural capabilities. Consistent with this, the size and/or neuroanatomical features of the cerebellum have been frequently linked with various behavioural or ecological strategies in mammals[4–6], birds[4,7–11], fish[12,13], amphibians[14,15] and squamate reptiles[3,16], but the poor shape correspondences of endocasts in the hindbrain region[17–20] as well as the taxon-specific or limited sampling in brain studies have typically hampered precise assessment and correlation analysis[21–29]. Furthermore, comparative studies have typically focused on identifying the influence of cerebellum volume on various ecological characters such as nest complexity, locomotor ecology, and/or prey capture behaviour, but additional volume-independent morphological parameters such as cerebellar foliation pattern and size could also be correlated[30,31]. As a result, despite the long history of volumetric assessment of individual brain subdivisions and stereological evaluation of neuroanatomical structures in vertebrates, quantitative evidence of cerebellar ecomorphology have so far relied on weight, linear dimension, or foliar measurements of freshly dissected or sectioned brains[30–34]. Hence, the extent to which interspecific cerebellar variations are reflected in terms of morphology, gene expression pattern, neuronal parameters, such as size, number, and spatial organisation and ecological behaviour is unclear. The cerebellum contributes to coordination, precision, and accurate timing of movement by receiving input from sensory systems of the spinal cord and from other parts of the brain, including the cerebral cortex. This structure is thus of key importance in the control of locomotion, a fundamental attribute to several ecologically crucial functions, such as feeding, predator avoidance and territorial defense. Thus far, comparative studies investigating locomotor adaptations within specific clades of vertebrates have largely focused on other functional and anatomical aspects such as limb morphology and performance, muscle type and physiology, axial skeleton morphology and proportion, as well as organisation of major descending motor pathways[35].

Here, we perform a comparative, integrative characterisation of cerebellum evolution in vertebrates, using a quantitative approach combining morphological, phylogenetic, ecological, volumetric, cellular, and transcriptomic data. We hypothesise that locomotor behaviour could be a strong predictor of cerebellar complexity at different levels of organisation, including volume, shape, neuron spatial arrangement, and gene expression pattern. To test this hypothesis, we use squamate reptiles—lizards and snakes—as the main model system because of their high levels of morphological diversity, including in the size as well as general anatomical and cellular organisation of the cerebellum[15,36–39]. Moreover, squamates exhibit unique ecological and behavioural features[35,40,41], and both the locomotor pattern and locomotor performance of snakes and lizards are surprisingly well-characterised and known to be strongly influenced by limb and body shape and/or length[35,42], as well as by several habitat attributes, such as temperature, substrate and inclination[43–45]. We assess cerebellar phenotypic diversification with respect to locomotor specialization in a representative dataset of squamate species, by integrating geometric morphometric and volumetric analysis of high-definition 3D brain models along with investigation and quantification of neuron distribution based on 3D imaging of cleared whole-cerebella and immunohistochemistry. Furthermore, to provide molecular insights into developmental processes and/or mechanisms underlying cerebellar diversification, we use a transcriptomic approach to test whether differences in the species expression profiles are shaped more by their phylogenetic relationships or by differences in locomotor behaviours. Altogether, our characterisation of the squamate brain uncovers multi-level variations in cerebellar structure. Along with changes in cerebellar shape and size, we show heterogeneity in Purkinje cell spatial distribution and dynamic gene expression pattern, which are all associated with locomotor specialization. These data indicate that locomotor pattern is a strong predictor of cerebellar architecture, and highlight the importance of mosaic brain evolution in generating brain patterns shared by groups of squamate species with behavioural similarities.

## Results and discussion

**Overall morphology of squamate brain.** To explore potential relationships between brain morphology and locomotor specializations, we used high-definition 3D reconstructions of whole-brains and isolated cerebella based on contrast-enhanced computed tomography (CT) and manual segmentation, using a representative panel of 40 lizard and snake species with different locomotor modes (Fig. 1 and Supplementary Table 1). Tissue fixation and staining procedures were performed using low concentrations of fixatives and contrast agents based on previous reports[46–51] and optimised protocols[52–55], thus avoiding or limiting soft tissue artifacts such as shrinkage (see Methods and Fig. 2). Furthermore, as observed in our study (Fig. 2a), brain structures have been shown to preserve their shape and cytoarchitecture following contrast enhancer staining[47,49–51], likely as a result to the small cell size and/or high neuron density. Major locomotor patterns were defined based on cranial and post-cranial anatomical features, habitat use, and movement type, which are all intimately associated with locomotor performance[35,40,42] and well-documented in quadrupedal, limb-reduced, and limbless squamates (see all descriptions of categorisation in Methods and Supplementary Table 1). For example, the limbless burrower group contains both lizard and snake species showing elongated bodies with an increased number (>26) of presacral vertebrae, reduced limb skeletal elements, and cylindrical skulls to reduce energy expenditure during burrowing[41,56], and at least four discrete types of locomotion have been recognised in snakes[35]. In addition to covering all major squamate groups, our dataset includes nine different species of scincid lizards, a geographically widespread family that displays a large array of post-cranial and limb morphologies, habitats, and locomotor behaviours (Fig. 1a and Supplementary Table 1). The wide heterogeneity in both morphology and spatial configuration of squamate brain subdivisions is already apparent

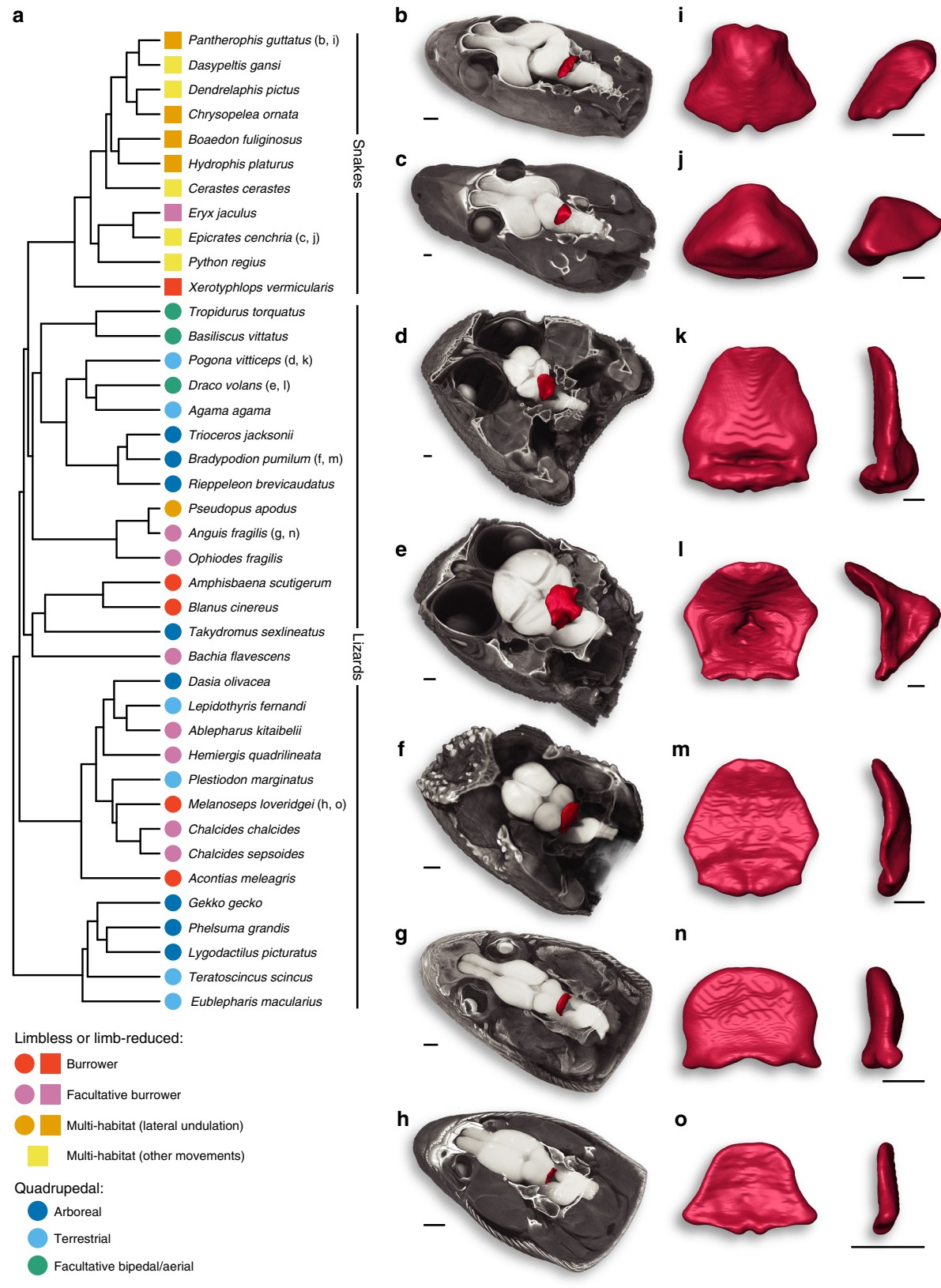

from a global inspection of our 3D models (Fig. 1b–h). Especially, the shape, size and degree of development of the cerebellum diverge extensively among species, ranging from a trapezoidal structure in snakes (Fig. 1i, j) to a more hexagonal conformation shared by quadrupedal lizards (Fig. 1k–m). Furthermore, snake and lizard cerebella generally display an opposite tilting relative to

brain anatomical axes, caused by a flexure occurring, with varying extents, either on pial (for snakes; Fig. 1i, j) or ventricular (for lizards; Fig. 1k–o) surface. As a result, whereas the snake cerebellum is caudally tilted (Fig. 1b, c, i, j), the lizard counterpart extends dorsally (Fig. 1k–o) and overarches the optic tectum towards the rostral end of the brain in quadrupedal species

**Fig. 1** Phylogeny and cerebellar diversity of squamates. **a** Phylogenetic tree of all snake and lizard species used in morphological and volumetric analyses, adapted from the most inclusive phylogenetic study available for extant squamates[85]. The seven major locomotor modes for snakes (coloured squares) and/or lizards (coloured circles), as defined based on anatomical features, habitat use, and movement type, are indicated by the same colour code throughout the entire manuscript (see bottom left corner): limbless or limb-reduced burrower (red squares and circles); limbless or limb-reduced facultative burrower (purple squares and circles); limbless or limb-reduced multi-habitat using lateral undulation (orange squares and circles); limbless or limb-reduced multi-habitat using other movements (yellow squares); quadrupedal arboreal (dark blue circles); quadrupedal terrestrial (light blue circles); quadrupedal facultative bipedal/aerial (green circles). **b–o** 3D-volume rendering and high-resolution whole-brain segmentation of iodine-stained adult heads (**b–h**) highlighting the cerebellum structure (**b–o**, red colour) of selected representative squamates at indicated position in the phylogenetic tree: *Pantherophis guttatus* (**b**, **i**), *Epicrates cenchria* (**c**, **j**), *Pogona vitticeps* (**d**, **k**), *Draco volans* (**e**, **l**), *Bradypodion pumilum* (**f**, **m**), *Anguis fragilis* (**g**, **n**), *Melanoseps loveridgei* (**h**, **o**). High magnifications of 3D-rendered cerebella (**i–o**) are shown in pial surface (left panels) and lateral (right panels) views for each selected species. Scale bars: 1 mm (**b–h**), 500 μm (**i–o**).

(Figs. 1d–f and 2a). Such inverted tilting is particularly pronounced when comparing similar cerebellar views of snakes (Figs. 1i, j and 2c) versus quadrupedal lizards (Figs. 1k–m and 2b).

**Brain shape diversity in squamates.** We next quantified the shape of the squamate brain, based on organ outline, as a whole and as a set of main subdivisions, including hindbrain subregions such as the cerebellum barely accessible on brain endocast[18,27], using 3D reconstructions and geometric morphometric approaches. Our landmark-based principal component analysis (PCA) performed on Procrustes coordinates (Supplementary Fig. 1) generated a morphospace defined by three first principal components—PC1 to PC3—which together account for more than 60% of the total shape variation both for the whole-brain (Fig. 3) and each individual subdivision tested (Fig. 4). Remarkably, whole-brain shape variations along the PC1 axis, from negative to positive values, reflect the morphological transition from snakes to quadrupedal lizards (Fig. 3). Indeed, PC1 negative values contain all snake species, which are characterised by laterally compressed optic tectum and compact forebrain showing stout olfactory bulbs and tracts as well as ventro-laterally expanded cerebral hemispheres (see, e.g., *Python regius*; Fig. 3). In contrast, apart from two chameleon species with stunted olfactory tracts that positioned at negative PC1 values (Fig. 3 and Supplementary Fig. 2b), all quadrupedal lizards are scattered along the positive PC1 axis, featuring antero-posteriorly compressed and laterally-protruding cerebral and tectal hemispheres as well as thin and elongated olfactory lobes and tracts (see, e.g., *Trioceros jacksonii*; Fig. 3). Interestingly, limbless and limb-reduced lizards populate the central part of the PC1 axis and display features ranging from a snake-like organisation in burrower lizards (see, e.g., *Melanoseps loveridgei*) to morphologies resembling that of quadrupedal lizards in facultative burrowers (see, e.g., *Chalcides chalcides*; Fig. 3). Lizards also display appreciable whole-brain morphological variations along the PC2 axis, which is associated with changes in the length and curvature of both olfactory tracts and medulla oblongata as well as in the reciprocal arrangement and dorso-ventral expansion of cerebral hemispheres and optic tectum (compare, e.g., *Bradypodion pumilum* versus *Xerotyphlops vermicularis*; Fig. 3).

Importantly, snake and lizard species tend to segregate in all our analyses of whole-brain and individual subdivisions, with snakes clustering at extreme positions along PC1 and/or PC2 axes, except for the cerebellum that displays an increased overlap based on two landmark configurations (Figs. 4 and 5a and Supplementary Fig. 1). Consistent with this pattern, and although a significant phylogenetic signal was identified for all subdivisions using a multivariate *K*-statistic[57] (*K*-values ranging from 0.5684 to 0.7118; *p*-values < 0.001; Supplementary Fig. 2a, b and Supplementary Table 2), the cerebellum was the only structure showing significant shape correlations with locomotor modes, as

assessed by phylogenetic ANOVA[58] (*p*-value = 0.0046; Supplementary Table 3). Notably, post hoc pairwise comparisons corrected for phylogenetic information indicate significant morphological differences between limbless burrowers and all other locomotor modes (phylogenetic ANOVA, *p*-values ranging from 0.0004 to 0.0216), as well as between any limbless or limb-reduced and quadrupedal locomotion (*p*-values ranging from 0.0001 to 0.0147; Supplementary Table 4). The segregation of limbless burrower species with other locomotor groups is conspicuous within cerebellum morphospace at negative PC1 values, which reflect a thin, triangular-shaped cerebellum with a barely detectable curvature (see, e.g., *Xenotyphlops vermicularis*; Fig. 5a). Along PC1 axis, cerebella progressively increase their dorso-ventral extension, expand laterally, and show a gradual intensification of the ventricular surface flexure (see, e.g., *Bachia flavescens* and *Phelsuma grandis*; Fig. 5a). At extreme PC1 positive values, they are remarkably developed along the dorso-ventral axis, with a considerable elongation of their medial regions that bend rostrally to overstep the tectal hemispheres (see, e.g., *Trioceros jacksonii*; Fig. 5a). Importantly, the PC2 axis distinguishes other limbless or limb-reduced locomotor modalities from quadrupedal locomotion, with cerebellar shape ranging from a dorso-ventrally stunted and laterally compressed structure typical of limbless or limb-reduced multi-habitat species and partially shared by some facultative burrowers at negative values (see, e.g., *Chrysopelea ornata* and *Chalcides chalcides*; Fig. 5a), to the dorsally and laterally developed and anteriorly-projecting cerebellum of quadrupedal lizards at positive values (see, e.g., *Phelsuma grandis*; Fig. 5a). Together with these changes, PC2 further marks the inversion of the cerebellar flexure from the pial surface at very negative values (see, e.g., *Python regius*; Fig. 5a) to the ventricular surface (compare, e.g., *Chrysopelea ornata* and *Chalcides chalcides*; Fig. 5a). Interestingly, the distribution of scincid lizards, which belong to four different locomotor categories, illustrates well the locomotor behaviour trends by their scattering throughout most of the cerebellum morphospace (see exact positions in Supplementary Fig. 2a). Furthermore, distance-based convergence measures from Stayton[59] support the significant cerebellar convergence of lizard and snake species within most of limbless or limb-reduced locomotor groups (*p*-values ranging from 0.0020 to 0.0360 in limbless burrowers and limbless or limb-reduced facultative burrowers; Supplementary Table 5).

The lack of significant locomotor signal on the whole-brain with phylogenetic ANOVA (*p*-value = 0.8076; Supplementary Table 3) might derive from the complex relationship between neuroanatomical structures and their function, with morphological changes likely reflecting several differences in behaviour and ecology. Importantly, similar cerebellar distribution patterns and shape changes were obtained using alternative landmark-free methods (Supplementary Fig. 2c, d), thus underscoring the robustness of our data. To better visualize and topographically map cerebellar shape

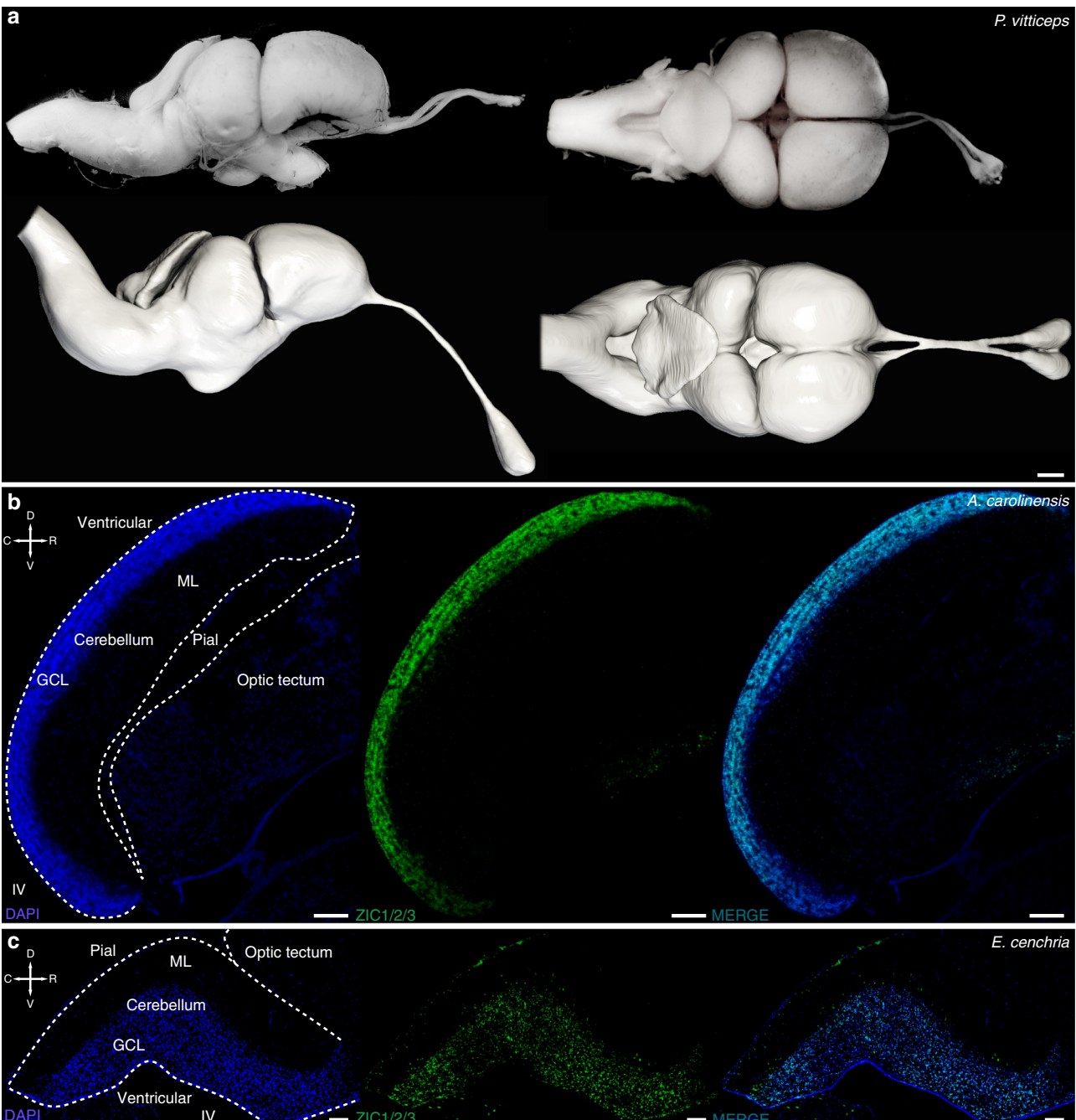

**Fig. 2** Comparative anatomy of whole-brain and cerebellar cortex. **a** Direct comparison of dissected (top panels) and 3D reconstructed (bottom) *Pogona vitticeps* brain in lateral (left panels) and dorsal (right) views. **b, c** Single and merged immunostainings for ZIC1/2/3 granule cell marker (green) and DAPI (blue) on sagittal sections of *Anolis carolinensis* (**b**) and *Epicrates cenchria* (**c**) cerebellar cortex. White dashed lines highlight both the cerebellum profile and part of optic tectum in contact with the cerebellum to illustrate their spatial relationships in the two species. Crossed white arrows point towards rostral (R), caudal (C), dorsal (D) and ventral (V) directions. Pial and ventricular cerebellar surfaces are indicated on their respective side. GCL, granule cell layer; ML, molecular layer; IV, fourth ventricle. Scale bars: 1 mm (**a**), 100 μm (**b, c**).

changes typical of each locomotor mode, we next reconstructed the mean cerebellar shape configuration for each locomotor group (Fig. 5b). Consistent with a significant correlation between locomotion and cerebellar morphology, considerable deviations from the overall mean shape for the entire dataset are apparent in specific areas for most locomotor groups. Especially, a large heterogeneity exists on both ventricular and pial surfaces, where bending and thickening diverge among locomotor groups, as well as in medial and lateral peduncular areas (Fig. 5b). For example, the pial surface curvature exhibits the highest degree of variation in

limbless or limb-reduced multi-habitat species, whereas limbless or limb-reduced burrowers, and to a lesser extent facultative burrowers, rather show major changes on the ventricular side. Similarly to other vertebrates, the reptile cerebellum is well-known to be divided into an elaborate array of regions that form an exquisitely organised topographic map regulating sensory-motor behaviour[2,3]. Especially, the medial and lateral parts of the main cerebellar body in snakes and lizards, two regions that show major changes in limbless or limb-reduced species from our dataset, have already been proposed to control axial musculature and limb

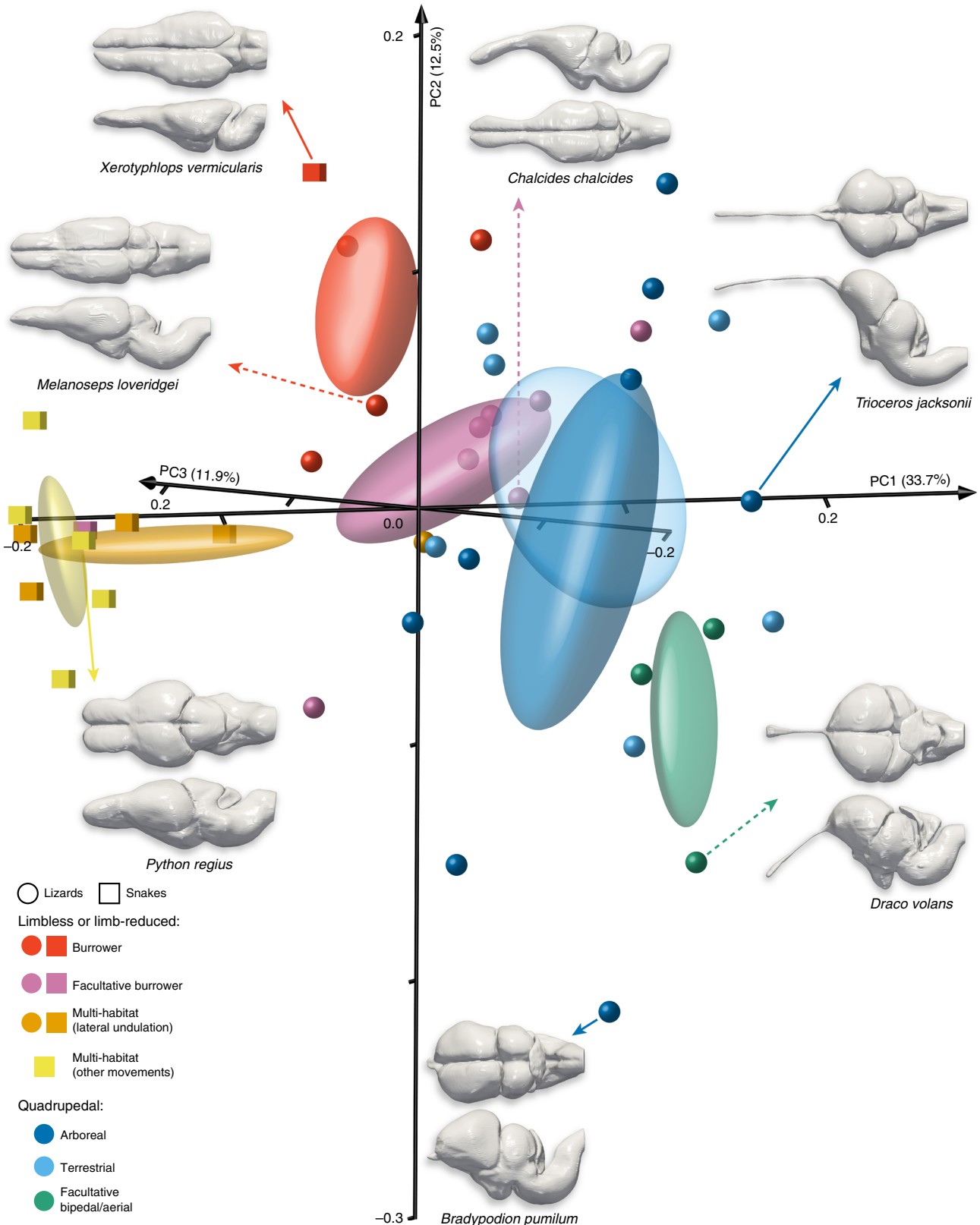

**Fig. 3** Whole-brain shape variation in squamates with different locomotor behaviours. 3D plot of principal component (PC) scores showing the whole-brain shape distribution of snakes (coloured cubes) and lizards (coloured spheres) with different locomotor modes (see colour code and symbols in bottom left corner). Numbers in brackets indicate the percentage of variance explained by each of the PC axes, and 68% confidence ellipses are shown for each locomotor mode. The 3D-rendered whole-brains corresponding to extreme (indicated by coloured solid arrows, with species names) or representative (coloured dashed arrows, with species names) lizard and snake species in both positive and negative directions along the first two PCs are shown in dorsal (top) and lateral (bottom) views. Source data are provided as a Source Data file.

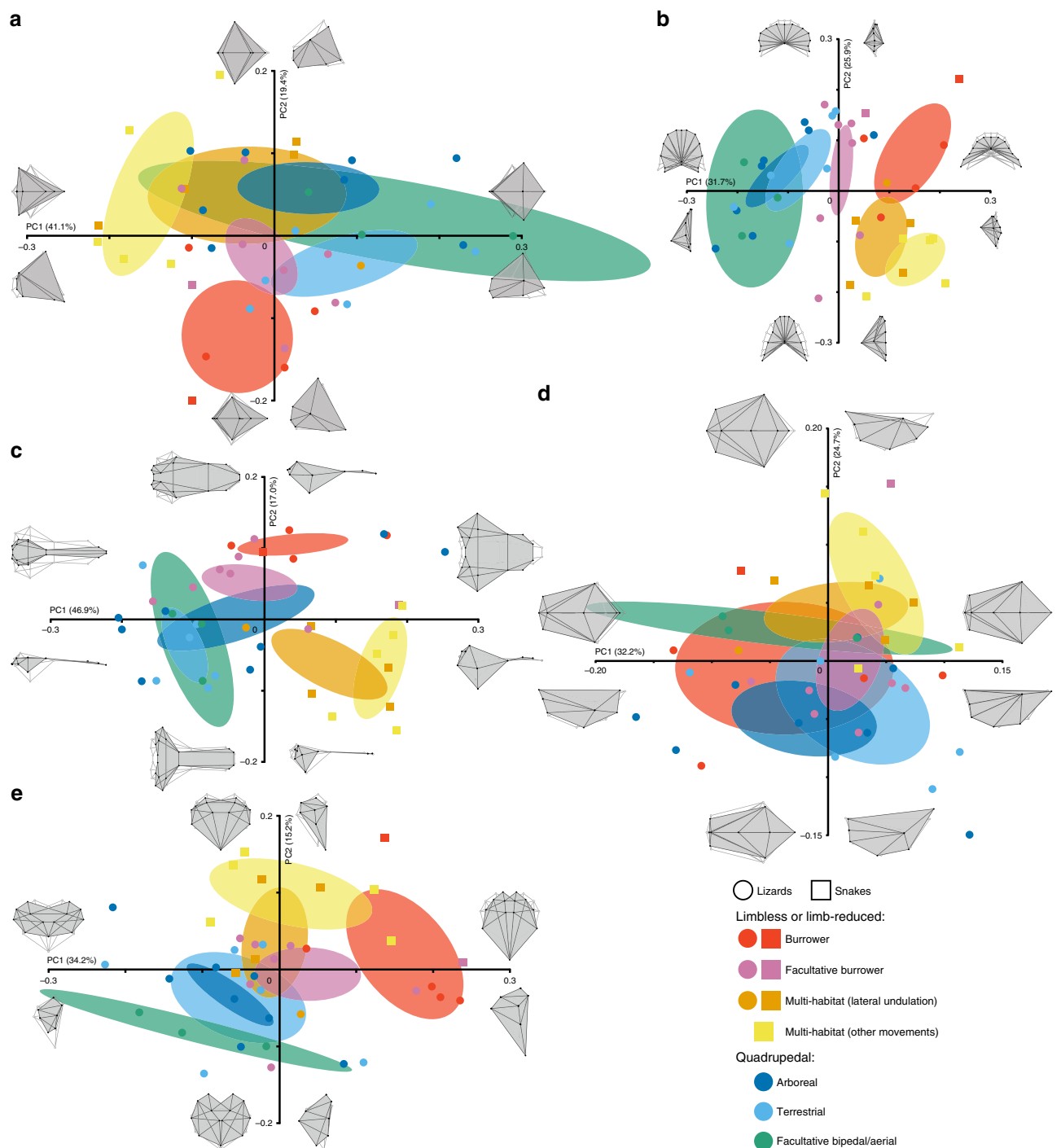

**Fig. 4** Morphological variation of major brain subdivisions in squamates with different locomotor behaviours. **a–e** 2D plots of principal component (PC) scores showing the diencephalon (**a**), cerebellum (**b**), telencephalon (**c**), medulla oblongata (**d**) and mesencephalon (**e**) shape distribution of snakes (coloured squares) and lizards (coloured circles) with different locomotor modes (see colour code in bottom right corner). Numbers in brackets indicate the percentage of variance explained by each of the PC axes, and 68% confidence ellipses are shown for each locomotor mode. 2D wireframes illustrate shape changes associated with the first two PCs for each brain subdivision in top and lateral (**a**, **c**, **d**), pial surface and lateral (**b**), or front and lateral (**e**) views. Source data are provided as a Source Data file.

movement[2,3], respectively. Furthermore, the presence/absence and degree of development of motor control brainstem nuclei and spinal pathways, including the red nucleus, vestibular nuclei, certain reticular nuclei, as well as the rubro-spinal, vestibulo-spinal, vestibulo-reticulo-spinal, and reticulo-spinal tracts, have been associated with the different body plans and locomotor behaviours of reptiles[60–64], but also with their level of cerebellar

complexity[60,61,64]. Our observed cerebellar shape changes are thus definitely expected to functionally correlate with both locomotor system anatomy and locomotor behaviour.

**Cerebellum size variability in squamates.** Because of the well-documented impact of ecological parameters on the relative size of

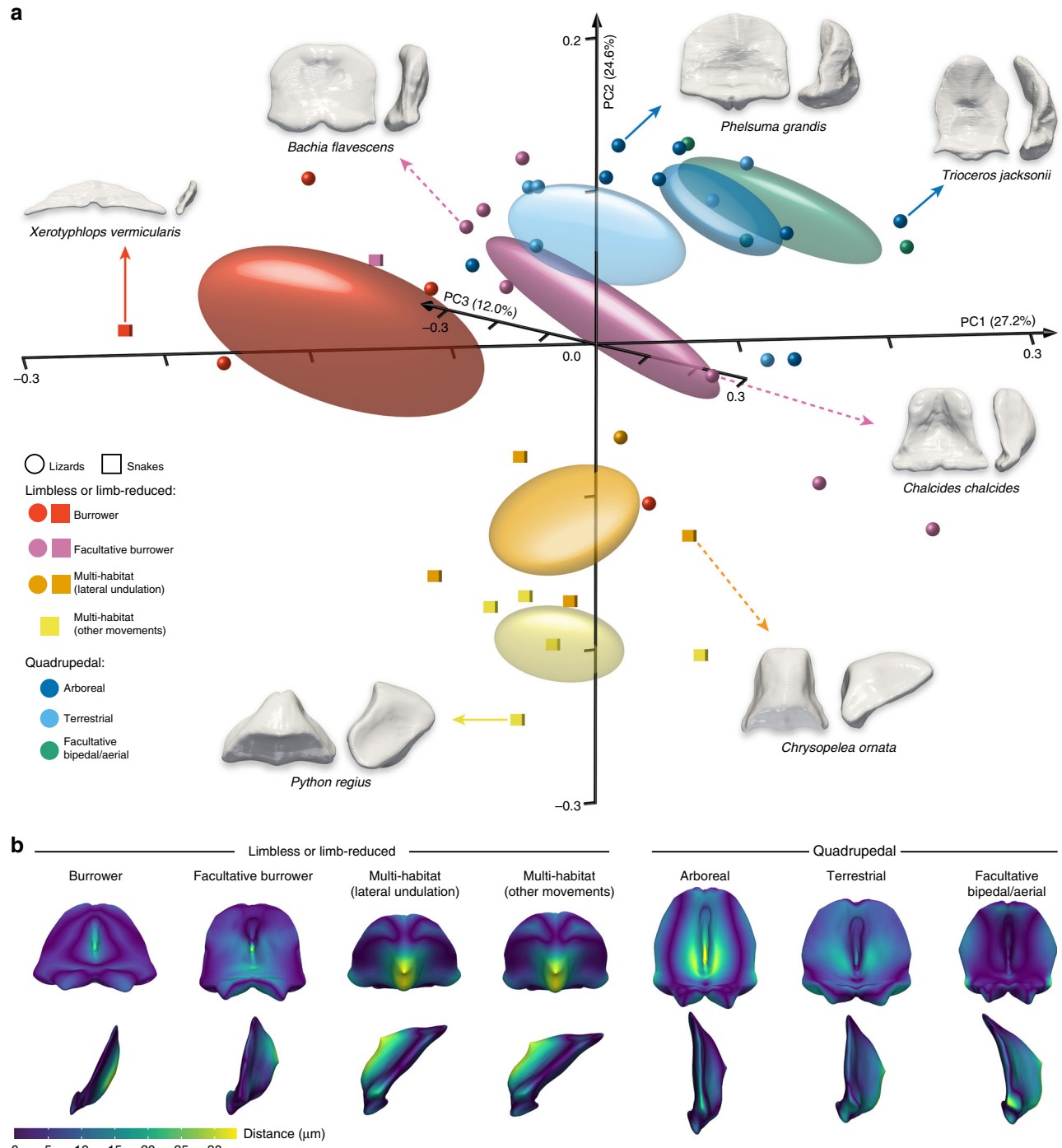

**Fig. 5** Cerebellar shape variation in squamates with different locomotor behaviours. **a** 3D plot of principal component (PC) scores showing shape distribution of isolated cerebella from snakes (coloured cubes) and lizards (coloured spheres) with different locomotor modes (see colour code and symbols in bottom left corner). Numbers in brackets indicate the percentage of variance explained by each of the PC axes, and 68% confidence ellipses are shown for each locomotor mode. The 3D-rendered cerebella corresponding to extreme (indicated by coloured solid arrows, with species names) or representative (coloured dashed arrows, with species names) lizard and snake species in both positive and negative directions along the first two PCs are shown in pial (left) and lateral (right) views. Source data are provided as a Source Data file. **b** Warped surfaces of cerebellum representing the reconstructed mean shape configuration for each indicated limbless or limb-reduced (left panels) or quadrupedal (right panels) locomotion mode are shown in pial surface (top row) and lateral views (bottom). Colour gradient, ranging from blue to yellow, reflects the relative Procrustes distance (in μm) of shape changes from the overall mean shape for the entire dataset.

brain subdivisions in most vertebrate lineages[6,12–16,21,23,25–28,32–34], we next compared the volume of our 3D cerebellar models among locomotor groups. In parallel with the significant effect of evolutionary allometry on cerebellar shape (multivariate regression analysis of Procrustes distance on centroid size, percentage predicted = 12.6%, p-value < 0.0001), our volumetric data highlight substantial divergence in cerebellar size both in absolute and relative terms (Fig. 6a). As the homogeneity of regression slope assumption

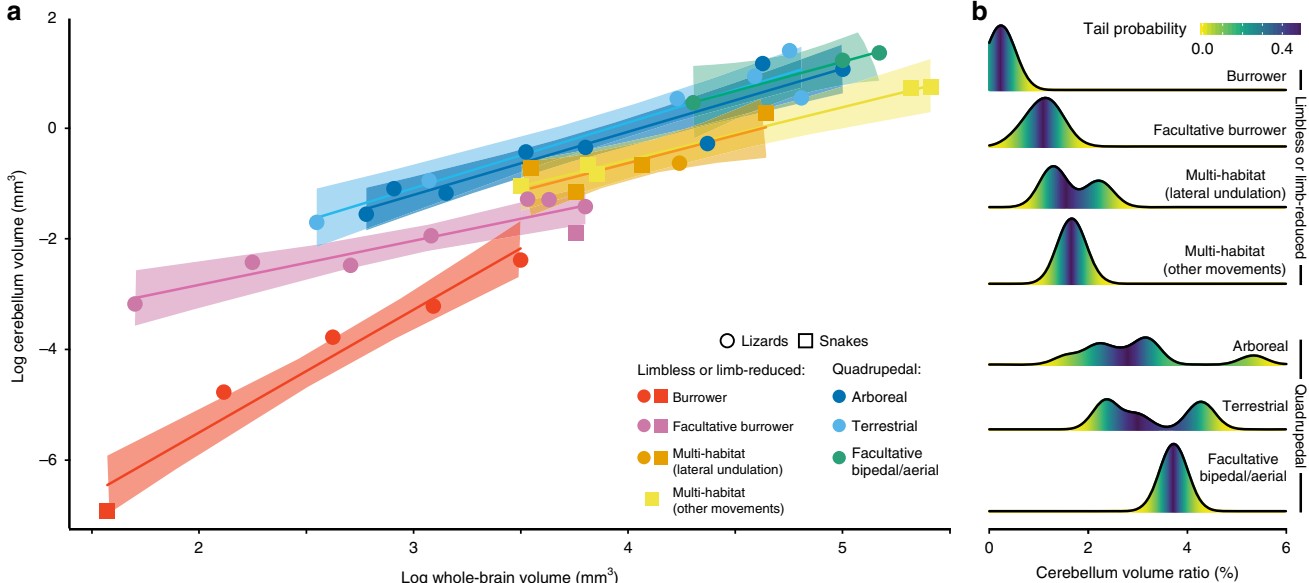

**Fig. 6** Cerebellar size variation in squamates with different locomotor behaviours. **a** Scatter plot showing the correlation of cerebellum relative to whole-brain volume (in mm$^3$) for selected representative squamate species (see colour code and symbols in bottom right corner). The coloured lines and shadings represent the phylogenetic generalized least squares (PGLS) regression lines and 95% confidence intervals for each locomotor mode, respectively. **b** Ridgeline plot showing the distribution of the cerebellum-to-whole-brain volume ratios (in percentage) for each indicated limbless or limb-reduced (top panels) or quadrupedal (bottom) locomotor mode. Colour gradient, ranging from yellow to blue, reflects the tail distribution probability.

required for conventional phylogenetic analysis of covariance (ANCOVA) was violated in our dataset (Supplementary Table 6), we used the alternative Johnson-Neyman procedure[65] to compare the locomotion-specific regression lines by establishing region of significance among volume covariates. Although there is not yet a phylogenetically informed implementation of the Johnson-Neyman method, the major assumptions of this technique are similar to ANCOVA, and the negligible phylogenetic signal (equivalent to zero) observed in our phylogenetic ANCOVA analysis allowed us to confidently use this method. Unexpectedly, this approach revealed few significant differences in the relative size of cerebella between locomotor groups. Indeed, only limbless burrowers showed a significantly reduced cerebellum within a restricted range of low whole-brain values (log volume < 2.42 mm$^3$; Supplementary Table 7). Importantly, similar significant differences of limbless burrowers with all other locomotion groups were further confirmed using phylogenetic ANOVA analysis (p-values < 0.05) on the cerebellum-to-whole-brain volume ratio (Fig. 6b). Altogether, these results highlight the great phenotypic diversity of the cerebellum across squamates, and unambiguously indicate that both the cerebellar shape and size reflect locomotor behaviours. Brain size has been traditionally the preferred measured trait in past evolutionary studies, but we show here that size is only one metric, and other parameters such as shape can change independently of brain region volume. Particularly, our data clearly indicate that the cerebellar shape is likely a more relevant feature of vertebrate brain evolution than its relative size, as also suggested by recent adaptive radiation studies in several vertebrate lineages[66,67].

**Purkinje cell layout in the cerebellar cortex.** Histological and morphological evidence suggest that species-specific characteristics of the cerebellar architecture could also be linked to behaviour and/or sensory ecology in vertebrates[3,10,11,30,31,68], although this relationship is yet not well understood. To investigate this aspect in the squamate cerebellum, we first compared the complexity and general organisation of the principal layers and cell types of the cerebellar cortex, using histology and

immunohistochemistry (IHC) methods. We particularly focused on comparing the spatial organisation of Purkinje cells (PCs), which is known to be either arranged in a monolayer or scattered in a few lizard and snake species, respectively[3,36–39]. Our comparative examination of the two predominant cerebellar neuron types—PCs and granule cells (GCs)—with specific markers such as CALB1 and ZIC1/2/3, respectively, corroborates previous observations and indicates heterogeneity in PC spatial arrangement among squamate species (Figs. 2b, c and 7). Particularly, 3D light-sheet fluorescence microscopy imaging of cleared whole-cerebella indicates a scattered organisation of PCs throughout the molecular layer (ML) in *Boaedon fuliginosus* snakes, in contrast to the orderly monolayer of *Pogona vitticeps* lizards (Fig. 7a, b), thus confirming the atypical PC pattern of snakes[3,36,37]. To investigate further this phenotype in a larger squamate dataset, we explored the potential relationships between cell arrangement and squamate locomotor specialization, by quantifying the scattering of individual PCs with a numerical approach integrating IHC, image processing, and statistical analysis. We particularly focused on comparing groups of individuals belonging to different selected squamate species with specific locomotor modes. Post hoc pairwise comparisons following highly significant Kruskal-Wallis test (p-value < 0.0001) revealed four major groups reflecting significantly different PC distribution patterns within the ML (Fig. 7c): ordered monolayer (group I), ordered multilayer (II), scattered multilayer (III) and totally scattered (IV). The lack of significant segregation between some locomotor modes, including between burrower and facultative burrower species that all cluster in group II, might be due to species sampling limitations, especially the difficulty to get fully burrower species. However, the low number of species for some locomotor modes does not affect the overall significance of our data, as significantly different PC organisation was further confirmed by directly comparing the obtained four groups I–IV with similar Kruskal-Wallis statistical test (p-values < 0.0001). Interestingly, and consistent with our cerebellar shape and volumetric data, all quadrupedal species cluster in a single group with similar ordered monolayer of PCs (group I). In contrast, significant differences were observed

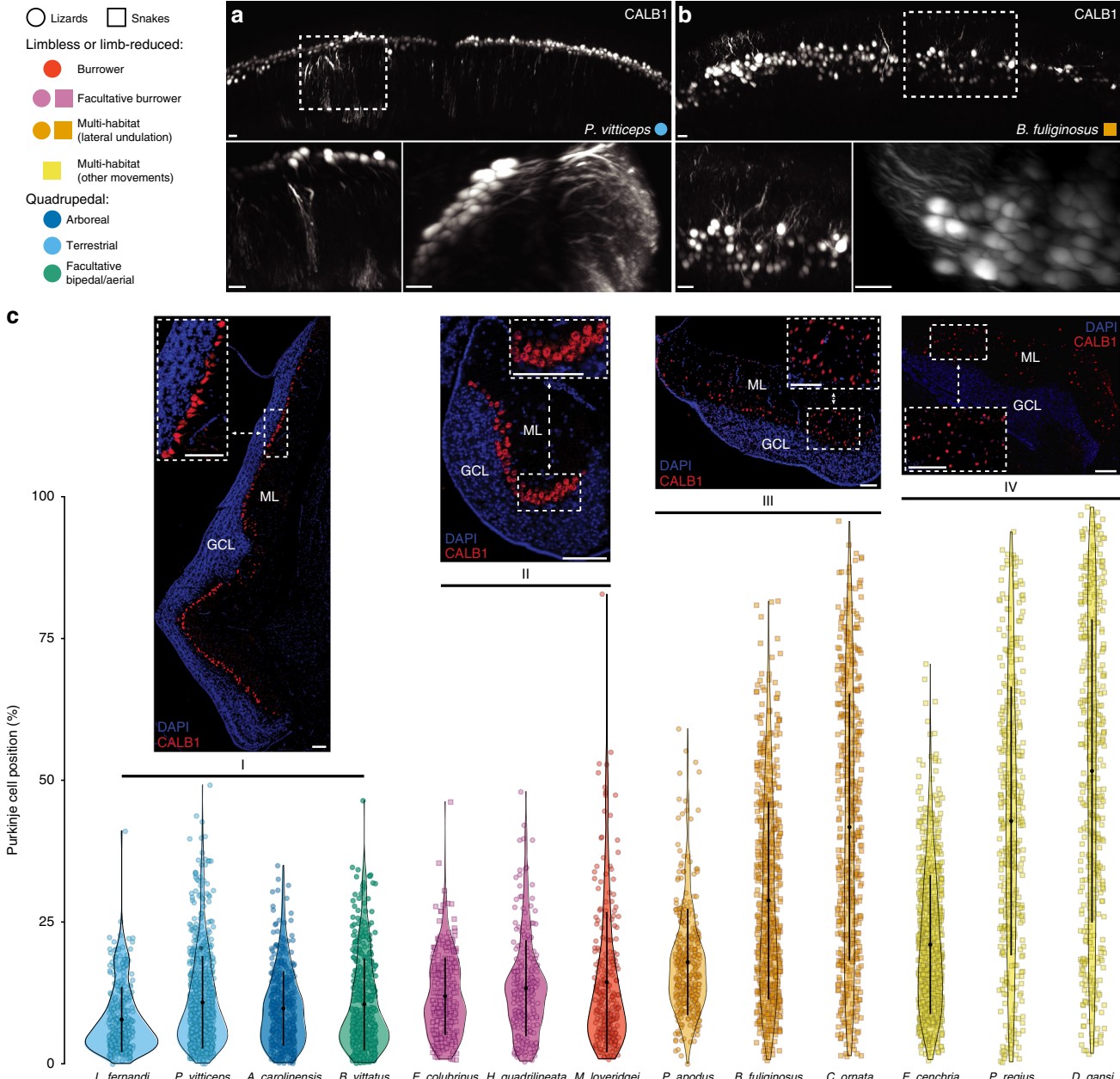

**Fig. 7** Variability in the arrangement of Purkinje cells (PCs) in the squamate cerebellum. **a, b** Representative light-sheet microscopy imaging of cleared whole-cerebella showing the 3D distribution and arrangement of calbindin 1 (CALB1)-immunolabelled PCs in two representative species with different locomotor modes (see colour code and symbols in top left corner): *Pogona vitticeps* (**a**) and *Boaedon fuliginosus* (**b**). The boxed areas in the coronal 3D-rendered cerebellar views (top panels) are shown at higher magnifications in coronal (left) and sagittal (right) views in the lower panels. **c**, Violin plot showing the quantitative distribution of CALB1-immunolabelled PCs in the cerebellar cortex of selected squamate species with similar or different locomotor modes (colour code and symbols as above). Due to intra- and interspecies heterogeneity in molecular layer (ML) thickness, the position of individual cells (*n* = 250–750 per species) was calculated as the distance (in %) from the granule cell layer (GCL) to the outer border (pial surface) of the ML, and error bars represent the standard deviation. Four major positioning patterns containing three or four squamate species and reflecting the increased scattering of PCs (from I to IV) were identified based on Kruskal-Wallis statistics. Immunohistochemistry with CALB1 marker (red staining) on sagittal sections of the cerebellar cortex in selected representative species are shown at low and high magnifications (insets) for each pattern: *Pogona vitticeps* (I), *Eryx colubrinus* (II), *Pseudopus apodus* (III), *Dasypeltis gansi* (IV). Source data are provided as a Source Data file. Scale bars: 30 µm (**a, b**), 100 µm (**c**).

among limbless and limb-reduced locomotion modes. The most striking example include the significantly different PC distribution pattern between groups that include both lizard and snake species, including burrower or facultative burrowers (group II) and multi-habitat species using lateral undulation (group III; Fig. 7c), suggesting that different degrees of PC scattering present both in snakes and lizards parallel locomotor specialization independently of species relationship. Altogether, our data indicate that the evolution of different locomotor strategies might require different kinds of cerebellar mediated coordination, resulting in a particular topological organisation of PCs likely different from the ectopic PC pattern previously reported in mutant mice or human patients with abnormal cerebellar development and neurological disorders[69–72]. Furthermore, future studies comparing neuron number and size as well as neuronal morphology, physiology, and complexity across vertebrate species

should reveal whether additional levels of cerebellar complexity are also ecologically relevant.

**Patterns of cerebellar gene expression.** Differences in gene regulation and expression pattern have long been recognised as crucial contributors to phenotypic diversity of the nervous system at both cellular and morphological levels[73]. Particularly, the accurate characterisation and quantification of orthologous transcripts across species are critical for understanding gene expression patterns and transcriptome-phenotype relationships, as evolutionary changes in gene expression contribute to behavioural phenotype and play a key role in phenotypic changes between species. Here, we employed comparative transcriptomics to determine the similarity of cerebellar gene expression across representative squamate species with similar or different locomotor behaviours. We restricted our analyses to a set of 630 protein-coding genes that could be identified as one-to-one orthologs across all 10 species RNA sequencing data (Fig. 8a). The difficulty to recover shared orthologs in our lizard and snake dataset could be justified by the lack of genome data information in most analysed models as well as by the abundance of rapidly-evolving genes showing accumulation of amino acid changes in squamates[74,75], two factors that are known to affect the sensitivity of ortholog identification. Heat map analysis and hierarchical clustering based on normalised expression values of orthologous genes revealed three clusters with characteristic gene expression patterns, including similar (up- or downregulated, clusters 1 and 2) or differential (cluster 3) expression among squamate species (Fig. 8a). We further investigated the biological function of sets of genes characterising each identified cluster, using a functional enrichment analysis of Gene Ontology (GO) categories. Interestingly, whereas the GO distribution was very similar in the three different clusters for the first three high-level biological process terms, including cellular process, biological regulation, and developmental process, significant gene category enrichment was observed for more specific terms. Especially, cluster 3, representing differentially expressed genes, is highly enriched with genes associated to development (three enriched categories, corresponding to 105 genes) but also to locomotory behaviour (29 genes; Fig. 8b), indicating that functionally relevant genes can exhibit dynamic expression ranges across squamates. Importantly, most of the identified enriched genes linked to locomotory behaviour are already known to be predominantly expressed in the brain and to play a key role in motor coordination, balance, and/or locomotor activity in multiple vertebrate models such as zebrafish (e.g., Spatacsin), mice (e.g., Contactin-2, Astrotactin-1, Ephrin type-A receptor 4, Striatin, Alsin, Metabotropic glutamate receptors 1 and 5, Nuclear receptor coactivator 2, Hamartin), rats (e.g., Unconventional myosin-Va), and humans (e.g., Pumilio homolog 1). Among examples, knock-out mice for Contactin-2, a cell adhesion molecule expressed in the Purkinje fiber network and critical for neuronal patterning and ion channel clustering, exhibit severe ataxic phenotype consistent with defects in the cerebellum[76], and mice that lack Astrotactin-1 have abnormal development of Purkinje cells associated with defects in balance, coordination, and walking behaviour[77]. Our data thus suggest that the evolution of different locomotor strategies might require expression level modulation of the identified cerebellar genes, indicating that differential gene expression might reflect the alternative locomotor patterns in squamates. By contrast, housekeeping and structural genes such as regulators of cytoskeletal actin, as well as GO categories not directly or exclusively linked to the specific movement from place to place of an organism (as defined for the locomotory behaviour category), are significantly over-represented in clusters 1 and 2 (Fig. 8b), thus

also confirming the validity of our data and experimental design. To test whether differences in the species expression profiles are linked to phylogeny or locomotor behaviour, we next performed hierarchical clustering on pairwise correlation to build a tree-like structure (dendrogram) where species are clustered into hierarchies according to their degree of expression pattern similarity. Strikingly, our expression phylogeny from the set of orthologous genes identified in cluster 3 (but also of all orthologous genes) clearly differs from the molecular phylogeny expected for squamates, and rather reveals a locomotion-dominated clustering (Fig. 8c). Furthermore, the majority of the groupings have relatively high bootstrap support, thus strongly indicating that lizard and snake species clustered according to locomotor modes rather than phylogenetic relationships (Fig. 8c). Notably, the variable expression patterns between locomotor groups precisely correlate with the phenotypic changes observed at both cellular and morphological levels, supporting the existence of tight functional links between gene expression, cerebellar architecture, and locomotor behaviours.

**Concluding remarks.** Altogether, our characterisation of the squamate brain uncovers multi-level variations in cerebellar architecture as well as unique relationships between a major ecological behaviour and organ specialization in vertebrates. Along with significant changes in cerebellar shape and size across squamate species, our study indicates a peculiar heterogeneity in cortical neuron spatial distribution and gene expression pattern, which are all intimately associated with locomotor specialization. These data unveil the existence of several relationships between brain complexity and locomotion in vertebrates, indicating that locomotion mode is a strong predictor of cerebellar size, shape, PC spatial organisation, and gene expression levels. Furthermore, this also suggests that major behavioural transitions in vertebrate locomotion patterns are evolutionary correlated with key changes in cerebellar structure, in addition to morphological changes expected in structures such as limb and post-cranial skeleton. In this context, it would be very interesting to assess additional levels of cerebellar complexity across vertebrate species, including neuron number and size as well as neuronal morphology, physiology, and circuit complexity. Importantly, the observed correlation between cerebellar shape and ecological behaviour in our geometric morphometric analysis across multiple brain regions demonstrates the existence of brain patterns shared by groups of species with lifestyle similarities, in a similar way to 'cerebrotypes' initially described for mammals based on size/volume data[78]. Furthermore, these findings on morphological features indicate that the main brain subdivisions in squamates are not uniform structures but rather evolved independently, thus supporting a mosaic model of brain evolution, at least with respect to locomotor capacities, as previously reported in different vertebrate groups[28,79–83]. Altogether, our work provides a framework for the evolution of cerebellar structure and locomotor behaviours in vertebrates, which can be reinforced by future experimental works and dissection of molecular and developmental mechanisms. In addition, our complementary analyses highlight how future comparative vertebrate studies of organ system-ecology relationships could benefit from adopting a multi-level integrative approach.

## Methods
**Data collection and categorisation.** Forty squamate species (29 lizards and 11 snakes) were sampled from collections at the Finnish Museum of Natural History (Finland), reptile colonies at the University of Helsinki (Finland), collaborators (Michael D. Shapiro, University of Utah, U.S.A.), and specialised retailers. All reptile captive breedings and experiments were approved by the Laboratory Animal Centre (LAC) of the University of Helsinki and/or the National Animal Experiment Board (ELLA) in Finland (license numbers ESLH-2007-07445/ym-23,

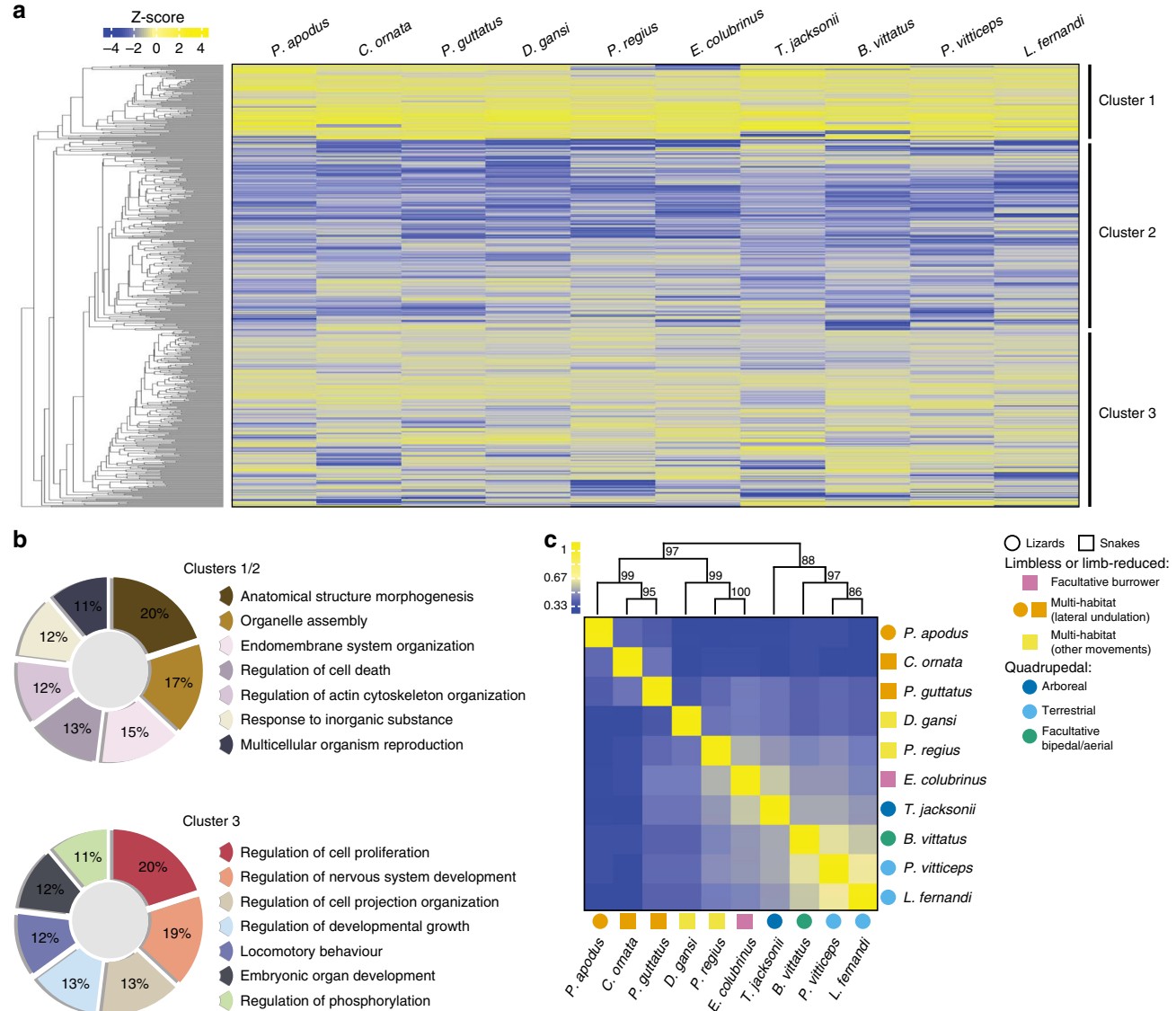

**Fig. 8** Comparative transcriptomics of the squamate cerebellum. **a** Two-way hierarchical clustering heat map showing three clusters of genes (rows) that behave similarly (clusters 1 and 2) or differently (cluster 3) across ten selected squamate species (columns). Z-score colour intensities reflect scaled gene expression values, ranging from low (blue) to high (yellow), for 630 one-to-one orthologous genes identified in all species. Source data are provided as a Source Data file. **b** Pie charts showing the distribution of orthologous genes (in %) among all significantly enriched gene ontology terms for biological processes (hypergeometric test with a false discovery rate multiple-hypothesis correction, $p$-value < 0.01) in clusters identified in **a**. **c** Hierarchical clustering of pairwise Pearson's correlation coefficients for 630 orthologous genes identified across all squamate species. Colour intensities of individual tiles in the heat map depict pairwise correlation coefficient values, ranging from low (blue) to high (yellow), between selected species with indicated locomotor mode (see colour code and symbols on the right). Numbers at nodes in the cluster dendrogram represent approximately unbiased $p$-values (in percentage) obtained by multiscale bootstrap resampling.

ESAVI/7484/04.10.07/2016, and ESAVI/13139/04.10.05/2017). *Hemiergis quadrilineata* specimens were collected under Licences to Take Fauna for Scientific Purposes from the Western Australian Department of Conservation and Land Management (No. SF003009 to Michael D. Shapiro) and exported under Environment Australia (EA) Permit to Export (No. PWS 994167; EA Approved Institution Number AI1146). All specimens analysed were at young adult stage (8–18 months after hatching), and head length (HL) reflecting the size of individuals was measured from the tip of the snout to the atlanto-occipital junction[84] (Supplementary Table 1). An average of 2.6 individuals per species was examined (range 1–8 depending on species sampling difficulties and number of analyses performed; Supplementary Table 1), and specimens were used for multiple observations and/or analyses whenever possible. Species were carefully selected to cover all major groups of squamates[85], thus enabling a comprehensive representation of the diversity of locomotor behaviours (Supplementary Table 1). Categorisation of locomotor groups was performed by cross-correlating data from reptile databases, available literature, and personal observations on different criteria: cranial and post-cranial anatomical features as previously defined[41,86,87],

including overall skull shape, limb characteristics (intact: well-developed limbs; reduced: limbs with skeletal elements reduced in size, lost, or fused; vestigial: retention of rudimentary limb skeleton; limbless: absence of external limbs), and degree of body elongation (elongated body: increased number (>26) of presacral vertebrae); habitat modes (aquatic: adapted for aquatic life, including marine environment; terrestrial: adapted for surface locomotion and foraging, including saxicolous; burrower: adapted for digging, living primarily and foraging underground; facultative burrower: terrestrial with some leaf-litter or burrowing behaviour; arboreal: adapted for locomotion between tree branches or bushes; semi-arboreal: often inhabiting and frequenting trees but not completely arboreal; aerial: arboreal with aerial behaviour, including gliding); movement types associated with locomotor performance and species-specific locomotor typologies based on previous descriptions[35,41,42] (lateral undulation: limbless locomotion involving waves of lateral bending being propagated along the body; rectilinear: limbless locomotion consisting of movement in a straight line; sidewinding: limbless locomotion similar to lateral undulation in the pattern of bending, but differing in static contact with the substrate and lifting of the body; arboreal concertina: limbless locomotion

consisting of gripping with portions of the body while pulling or pushing other sections in the direction of movement; modified concertina: concertina movement involving active pushing against the sides of the tunnel to provide a static anchor; quadrupedal arboreal: locomotion using four limbs to move through trees, including morphological specializations; quadrupedal terrestrial: surface locomotion using four limbs; facultative aerial: locomotion occurring in the air, including gliding; facultative bipedal: form of terrestrial locomotion using the two rear limbs, including running). Seven major locomotor groups were defined based on these criteria, including two major categories (limbless or limb-reduced and quadrupedal) further divided into four (burrower, facultative burrower, multi-habitat lateral undulation, multi-habitat other movements) or three (arboreal, terrestrial, facultative bipedal/aerial) subgroups (see Supplementary Table 1): limbless burrower (burrower species using modified concertina or rectilinear locomotion); limbless or limb-reduced facultative burrower (facultative burrower species using slow lateral undulation locomotion); limbless multi-habitat lateral undulation (terrestrial, aquatic, or aerial/arboreal species using lateral undulation locomotion); limbless multi-habitat other movements (terrestrial or arboreal species using rectilinear, arboreal concertina, or sidewinding locomotion); quadrupedal arboreal (arboreal species using quadrupedal arboreal locomotion); quadrupedal terrestrial (terrestrial species using quadrupedal terrestrial locomotion); quadrupedal facultative bipedal/aerial (semi-arboreal or aerial species using facultative bipedal or aerial locomotion).

**Generation of 3D whole-brain models.** High-resolution 3D CT scans of adult squamate heads were performed at the University of Helsinki or University of Kuopio imaging facilities (Finland) using Skyscan 1272 or 1172 (Brucker, Belgium). Samples were fixed and conserved in 10% formalin (museum specimens) or fixed in 4% paraformaldehyde (fresh specimens), two solutions that give almost identical final concentrations of formaldehyde (Supplementary Table 1). Previous quantitative analyses of brain tissues have demonstrated minor shape and cytoarchitecture artifacts introduced by such fixatives, including by comparing specimens with different preservation procedures[47,50,55], and limited shrinkage of brain tissues is expected because of the relatively low concentrations used[46,48–50]. For optimal brain tissue visualization, samples were next stained with 0.3% phosphotungstic acid (PTA) or 1% iodine solutions, two contrast enhancers with optimised protocol and known penetration power[52–55]. Based on previous reports[47–49,51] and own observations (see Fig. 2a), such low concentrations of contrast enhancers show either no or limited shrinkage of brain tissues, including in museum samples[51]. Scans were reconstructed using NRecon 1.7.0.4 software (Bruker) and 3D volume rendering as well as segmentation were done using the software Amira 5.5.0 (Thermo Fisher Scientific, U.S.A.). We adopted a manual segmentation approach to gain detailed brain reconstructions and to accurately reproduce the prominent traits of all encephalic subdivisions, including their spatial relationships (Fig. 2a). Most brain structures were discernible and could be included in the reconstructions, except for the pituitary and pineal glands. Both the whole-brain and isolated cerebellum were segmented, thus allowing assessment of volumetric and geometric morphometric measurements of these structures. Furthermore, geometric morphometrics was performed on each individual brain subdivisions using different specific subsets of landmarks on the whole-brain reconstructions (see below and Supplementary Fig. 1). A coronal plane intersecting the atlanto-occipital junction was set as the posterior limit of segmentation. The overall fixation/staining procedure and accuracy of all generated 3D models were carefully controlled by directly comparing our reconstructions with freshly dissected brains along the three anatomical planes (see Fig. 2a), and/or by incorporating more than one specimen as well as both museum and fresh specimens for each species whenever possible ($n = 1$–4 individuals per species depending on sample availability; Supplementary Table 1).

**3D whole-surface analyses and geometric morphometrics.** A total of 61 anatomical landmarks, clustered in groups delineating the 3D profile of the five major brain subdivisions (telencephalon, diencephalon, mesencephalon, cerebellum, medulla oblongata; Supplementary Fig. 1), were digitised on whole-brain surfaces using the Landmark Editor software (Institute for Data Analysis and Visualization, IDAV, U.S.A.). We further took advantage of our manual segmentation of the isolated cerebellum structure to improve its overall shape representation, by adding 5 extra-landmarks at the interface between mesencephalon and cerebellum, a region inaccessible on whole-brain models (Supplementary Fig. 1b). Measurement errors deriving from digitisation were estimated by repeating landmarking in two independent sessions (Supplementary Table 8), and obtained coordinate values were then averaged and used for subsequent analyses[88]. The accuracy of our 3D reconstructions and landmarks was further validated by controlling for shape outliers in the R-package geomorph v3.0.7[89] (https://CRAN.R-project.org/package=geomorph).

A Generalized Procrustes Analysis (GPA) was used to simultaneously superimpose all configurations and extract shape data by removing the effects of scale, orientation, and position[45]. Centroid size was computed as the square root of the sum of the squared distances of all landmarks from their centroid, and a multivariate regression of shape onto centroid size was used to test for allometry[90]. A Principal Component Analysis (PCA) performed on the covariance matrix of shape variables was used to visualize the main patterns of morphological variation

with 68.27% confidence ellipses for each group mean (5000 replicates), and the R-package pca3d v0.10 (https://CRAN.R-project.org/package=pca3d) was used to generate 3D PCA plots. Phylogenetic signal was calculated from shape data with a generalised K-statistic[47] in R-package geomorph v3.0.7[89], using the most inclusive and recent phylogenetic studies available for extant squamate species[85], and phylogenetic relatedness was corrected in subsequent statistical analyses to take into account independent observations across different species. To correct for allometry, all statistical analyses were also performed on residuals of the multivariate regression of shape onto size. The influence of locomotor modes on brain shape was tested with post hoc pairwise tests using phylogenetic ANOVA based on an advanced generalized least squares (GLS) approach[58] in geomorph package (function 'advanced.procD.lm'). Convergent evolution in the different locomotor groups was assessed using the distance-based convergence measures C1-C4[59] in the R-package convevol v1.3 (https://CRAN.R-project.org/package=convevol). To reconstruct the mean cerebellar shape configuration of each major locomotor mode, the closest specimen to the overall mean shape in multidimensional space was first identified via the function 'findMeanSpec' in geomorph package, and a 3D CT-scan of this specimen was warped to the overall mean shape (0.0) based on Thin Plate Spline (TPS) method[91] using the function 'tps3d' in R-package Morpho v2.6[92] (https://CRAN.R-project.org/package=Morpho). Patterns of shape changes were then visualized as deviations from this overall mean shape configuration to the averages of each locomotion group, using warping and heatmap functions from R-package Rvcg v0.18 (https://CRAN.R-project.org/package=Rvcg) and colour palette from R-package viridis v0.5.1 (https://CRAN.R-project.org/package=viridis). Phylogenetic ANCOVA was carried out with function 'phylol' from R-package phylolm v2.6[93] (https://CRAN.R-project.org/package=phylolm), using the most inclusive and recent squamate phylogeny[85]. The Johnson-Neyman procedure was used as an alternative to ANCOVA[65] when both heterogeneity of regression slopes and negligible phylogenetic signal (equivalent to zero) were observed, as this method currently lacks phylogenetically informed implementation. All analyses were performed in R with custom R scripts[94].

Our whole-brain reconstruction strategy allowed us to employ alternative landmark-free methods, including statistical particle-based models of shape (ShapeWorks)[95] and generalized Procrustes surface analysis (GPSA)[96], to corroborate the shape data derived from manual landmarking (Supplementary Fig. 2c, d). Default ShapeWorks parameters were used for the grooming step, whereas the following parameters were used for the optimisation step: number of particles: 2000; relative weight: 0; starting regularisation: 1000; ending regularisation: 10; iterations per level: 3000; decay span: 1000. For GPSA, reconstructed whole-brain surfaces were simplified to 100,000 vertices and converted to Polygon File Format (PLY) in Amira 5.5.0, before loading into the GPSA software. 3D between-group PCAs[97] for ShapeWorks and GPSA were used to visualize the main patterns of morphological variation among groups via function 'groupPCA' from R-package Morpho.

**Tissue clearing and light-sheet fluorescence microscopy.** Large portions of the hindbrain, containing the entire cerebellum as well as parts of the medulla oblongata and tegmentum metencephali, were dissected from adult *Pogona vitticeps* and *Boaedon fuliginosus* whole-brains ($n = 3$ individuals per species). Tissues were immediately fixed for 30 min at room temperature (RT) in 4% paraformaldehyde (PFA), dehydrated through a series of increasing methanol concentrations (25%, 50%, 75%, 100%) in phosphate buffered saline (PBS), and then stored at −80 °C. Prior to optical clearing, samples were subjected to five freeze-thaw cycles from −80 °C to RT to increase the permeability to antibodies, followed by rehydratation steps in a series of decreasing methanol concentrations (75%, 50%, 25%, 0%) in PBS. Hindbrain tissue clearing and whole-mount immunohistochemistry (IHC) were performed according to the Clear, Unobstructed Brain Imaging cocktails and Computational analysis (CUBIC) protocol[98], with minor modifications. PCs were labelled using primary anti-Calbindin D-28K antibodies (CALB1; 1:300, rabbit polyclonal, Swant, Switzerland, cat# CB38, RRID: AB_10000340) and Alexa Fluor-conjugated secondary antibodies (1:500, goat anti-rabbit IgG, Thermo Fisher Scientific, U.S.A., cat# A-11008, RRID: AB_143165). Cleared tissue fluorescence image-stack tiles were acquired with Zeiss Lightsheet Z.1 microscope (Zeiss, Germany), using 561 nm lasers and dual side illumination. Image-stack tiles were fused by applying the 'maximum intensity projection' algorithm from the ZEN software (Zeiss, Germany), and then aligned using Imaris Image Stitcher application (Bitplane, Switzerland). 3D-rendering and image acquisition of the reconstructed volumes were carried out in Imaris v9.2 software (Bitplane, Switzerland).

**Purkinje cell distribution quantification.** To quantify and compare the distribution of PCs in the cerebellar cortex of squamate species with different locomotor modes, IHC on paraffin sections covering one cerebellar half along the sagittal plane were selected in homologous positions along the medio-lateral axis in each specimen, because of the species-dependent heterogeneity in antibody penetration with whole-mount methods. IHC on sections was conducted according to standard protocol[84], using heat-induced epitope retrieval and overnight incubation at 4 °C with primary anti-CALB1 antibodies (see above) for specifically identifying PCs. The position of individual labelled PCs relative to the ML was calculated by

measuring the shortest distance between the center of PC somata and the inner border of the granule cell layer (GCL). Due to high intra- and interspecies heterogeneity in ML thickness that could affect the relative positioning of PCs, particularly in species with scattered PC organisation, all individual PC-GCL distances were then normalised to the entire ML thickness at particular PC location, by measuring the distance between the inner border of GCL and outer border of ML (pial surface). The particular anatomical localisation and organisation of the GCL, situated as a sharply edged and densely packed block of cells on the ventricular surface of the squamate cerebellum, allowed us to use 4′,6′-diamidino-2-phenylindole (DAPI) nuclear counterstaining (Sigma-Aldrich, U.S.A.) as a suitable proxy to outline the GCL. In addition, IHC with GCL-specific markers such as Zinc finger proteins 1/2/3 (ZIC1/2/3; 1:300, rabbit polyclonal, LifeSpan BioSciences, U.S.A., cat# LS-C118695) showed a consistent overlapping with DAPI labelling on the ventricular surface of the cerebellum in all tested species (Fig. 2b, c). Distance measurements were performed in the package Fiji[99] by running the 'measure distance to line' macro. As significant deviations from normality were obtained in the gathered data based on multiple methods (Shapiro-Wilk, Lillefors, Jarque-Bera, and Anderson Darling tests), thus precluding the use of parametric tests, the Kruskal-Wallis test followed by Dwass-Steel-Critchlow-Flinger post hoc comparative analysis was carried out to determine potential variations in PC distribution among locomotor groups ($n = 250–750$ observations (individual PCs) per species, using a range of 1–3 individuals from different species per locomotor mode depending on sample availability; Supplementary Table 1). Because of the low number of species for some locomotor groups, similar Kruskal-Wallis statistical analysis was also performed by combining some locomotor modes ($n = 3–4$ species per group in this case).Violin plots of PC distribution were obtained with function 'ggplot' from R-package ggplot2 v3.1.0 (https://CRAN.R-project.org/package=ggplot2).

**High-throughput RNA sequencing and transcriptome analyses**. Total RNA was extracted from freshly dissected cerebellar tissues of 10 different squamate species with different locomotor behaviours ($n = 1–3$ species per locomotor mode depending on sample availability; Supplementary Table 1), using the RNeasy Plus Micro Kit (Qiagen, Germany), according to the manufacturer's instructions. RNA sample concentrations were detected by Qubit 3.0 (Thermo Fisher Scientific, U.S.A.), and RNA Integrity Number (RIN) values were determined using Agilent 2100 Bioanalyzer (Agilent Technologies, U.S.A.). Libraries were produced using the TruSeq Stranded Total RNA Kit with Ribo-Zero (Human/Mouse/Rat; Illumina, U.S.A.), and paired-end sequencing ($88 + 74$ bp) was done using duplicates on a NextSeq 500 platform (Illumina, U.S.A.). Quality trimming and adapter sequence removal of reads were done using the Trimmomatic tool[100], and the SortMeRNA pipeline[101] was used for ribosomal RNA (rRNA) filtering. Quality trimmed and rRNA-filtered reads (total average of 44.2 million reads per species) were de novo assembled in contigs with the Trinity software v2.8.5[102], using its default parameters. The coding (CDS) and protein sequence of unigenes were then predicted using the TransDecoder program implemented in Trinity[102], and functional categories were determined using the Blast2GO program (BioBam, Spain)[103]. Finally, the OrthoFinder algorithm[104] was used for inferring orthologous and species-specific unigenes from protein sequences (total average of 8942 unigenes per species). The relative abundance of transcripts was evaluated for each species as FPKM (fragments per kilobase of transcript per million mapped reads) using the RNA-Seq by Expectation Maximisation (RSEM) software[105], and only orthologs expressed above a fixed expression threshold of 0.5 FPKM in at least one sample were retained to get a more accurate characterisation and quantification by excluding possible noise at very low expression levels. 630 one-to-one orthologs shared across all ten species (Source Data) were subsequently used for hierarchical clustering, correlation coefficient, and functional enrichment approaches. Expression heat maps and hierarchical clustering were generated with the Heatmapper software[106], using Euclidean distance or Pearson's and Spearman's rank correlation coefficients. The robustness of hierarchical clustering was further assessed using R-package pvclust v2.0-0[107] (https://CRAN.R-project.org/package=pvclust), with 1000 iterations of multiscale bootstrap resampling. Functional enrichment analysis was performed on several sets of orthologous genes based on hierarchical clustering, using the Gene Ontology (GO)-term enrichment tool available on GO Consortium website[108] (http://geneontology.org/) and R-package GOstats[109] (https://www.bioconductor.org/packages/release/bioc/html/GOstats.html). The significance of the enrichment was tested by the hypergeometric test with a false discovery rate multiple-hypothesis correction, and only significantly enriched GO terms ($p$-value $< 0.01$) comprising at least ten genes were considered.

**Reporting summary**. Further information on research design is available in the Nature Research Reporting Summary linked to this article.

## Data availability
Landmarks and main PC scores used for visualization and analysis are available in the main Figures and Supplementary Information. Illumina reads and processed RNA sequencing data have been deposited on the Gene Expression Omnibus (GEO) database under the accession number GSE139570. Source data on geometric morphometrics for Figs. 3–5, Purkinje cell counts for Fig. 7c, and gene expression for Fig. 8a are provided as

a Source Data file. 3D brain models are available through the corresponding author, upon reasonable request.

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

## Acknowledgements

We thank Ilpo Hanski and Martti Hildén (Finnish Museum of Natural History, Helsinki, Finland) for specimen loans; Michael D. Shapiro (University of Utah, U.S.A.) for collection and sharing of *Hemiergis* specimens; Arto Koistinen (University of Kuopio, Finland) and Heikki Suhonen (University of Helsinki, Finland) for access to X-ray computed tomography facilities; Petri Auvinen (University of Helsinki, Finland) for access to sequencing facility; Mika Molin and Kimmo Tanhuanpää of the Light Microscopy Unit (University of Helsinki, Finland) for optimisation of light-sheet microscopy imaging; the Office of Collaborative Science (OCS) microscopy core (NYU Langone Medical Center, U.S.A.) for the Fiji macro; Filipe Oliveira Da Silva for some CT-scans; Riikka Matikainen for technical assistance; members of the Di-Poï laboratory and Helsinki Evo-Devo community for discussions. This work was supported by funds from the Doctoral Program Brain & Mind (to S.M.), University of Helsinki (to N.D.P.), Institute of Biotechnology (to N.D.P.), and Academy of Finland (to N.D.P. and I.K.).

## Author contributions

S.M. and N.D.P. designed the overall experimental approach and selected the species sampling. Micro-CT scans and 3D reconstructions were carried out by S.M. S.M. collected the 3D landmark data. Y.S defined, implemented the geometric morphometric and statistical pipelines, and performed the geometric morphometric and statistical analyses. S.M. performed the expression pattern and cell quantification experiments. I.K., S.M., and N.D.P. analysed the transcriptome data. S.M. and Y.S made the figures, N.D.P and S. M. wrote the paper, and all co-authors contributed in the form of discussion and critical comments. All authors approved the final version of the paper.

## Competing interests

The authors declare no competing interests.
