## [Peer Review File · Nature Communications]

Reviewers' Comments:

Reviewer #1:

Remarks to the Author:

In this study the authors performed comparative morphological analyses of extant reptilian brains with multiple experimental strategies. The authors particularly focused on the Squamate cerebellum and executed 3-D computed tomography, whole mount imaging of Purkinje cell distribution, and transcriptomics in adult cerebellum, which conducted substantial associations among specific characteristics in these traits and various locomotive behaviors in limbless/limb-reduced or quadrupedal characteristics independent to phylogenetic relationship. The authors concluded that existence of significant correlations among these traits and locomotive behaviors resulted from convergent adaptive evolution, and that mosaic patterns of brain evolution can be categorized by cerebrotypes based on their definition. Each part of analysis had been performed in detail with enough number of samples and data representation were convincing, however, there are conceptual problems in the integration of the results and ambiguity in the purpose of the project.

- 1) The most significant problem is on lack of the causality among morphological traits and specific locomotive behaviors across species. It is unsure how unique shape of the cerebellum is functionally associated with the pattern of locomotion, Purkinje cell distributions, and transcriptome data represented in the manuscript. Without experimental proves or any relevant explanations for the functional connections, all the represented results are still fragmental and hard to conduct the principle underpinning the brain evolution.
- 2) Several previous comparative studies of locomotion and relevant neural circuits revealed significant associations between the nucleus ruber and reticulo-spinal tracts and quadrupedal behaviors in reptiles, while the cerebellum is relatively undeveloped in these animal groups, leading the question why the authors focused on the cerebellum but not examined other nuclear structures functionally associated with locomotive behaviors.
- 3) Adult brain morphology must be significantly affected by external physical constraint, i.e., the shape of skulls. For example, the length of olfactory tracts is tightly constricted by the morphology of the orbit in reptiles. The author have to elaborate how they could eliminate extrinsic influences to the brain shapes.
- 4) The project completely lacks the aspect of developmental biology that is critical to gain mechanistic insights into the evolution of species-specific body structures. "Developmental genes" in the adult tissue represented in the transcriptome data does not always reflect the functional association with embryogenesis, because many developmental genes are pleiotropic in their expressions and functions. The author must execute comparative ontogenic analysis of the cerebellum and cellular distributions in different species to clarify causal relationships among neuroanatomical architectures, gene expression, and unique locomotive behaviors.

Reviewer #2:

Remarks to the Author:

The authors present a unique and extended dataset of Squamata brains which allowed them to analyse the shape and size of the different regions. A clear correlation of the cerebellum shape is shown with the locomotion type of the species.

The manuscript is well written and illustrated and the information is clear. There are only a few points

that I believe should be addressed:

- provide a table with the number of individuals analysed for each type of analyses (3D model, histology, light sheet, RNA seq) as well as their age/size. It is important to show and maybe indicate in the text that only adult-sized animals were included in the study
- I am surprised by the small number of one-to-one orthologs, only 630 out of ~9000 unigenes. I believe that GO analyses are more meaningful when a larger set of genes is available, especially as one gene can belong to several categories. How many genes are shared by nine species and how different is the GO distribution of the genes in the three clusters?
- The brain is known to express a large number of genes where their function is often unknown. In cluster 3, the authors detect 12% of the genes (how many genes does this correspond to?) linked to locomotory behaviour. I believe it is important to discuss further these findings. Which genes are these? Why would their expression in the brain affect the locomotion of the animal? How relevant or random is this finding, compared for example to the 'response to inorganic substance' or 'multicellular organism reproduction' in clusters 1/2?

Reviewer #3:

Remarks to the Author:

The manuscript submitted by Macri et al. is a very ambitious, inter-disciplinary approach to examining brain evolution in a clade that has been the focus of relatively few studies: squamates. An impressive suite of approaches were brought to bear on cerebellar anatomy specifically and there are some interesting and very novel findings presented. However, the results are presented in a manner that I found extremely confusing and it was difficult to distill what conclusions could be made. The over emphasis on referring to the supplemental material detracted from the manuscript itself and the figures presented within the manuscript were difficult to understand. Adding to the confusion and interpretability was a densely written methods section that did not adequately explain why certain analyses and tests were done and introduced others that I do not think are appropriate in the context of phylogeny-informed analyses. I strongly recommend that the authors consider breaking this manuscript up into more than one paper or significantly expanding the current one so that the reader can understand what was done and why. I provide more detailed comments below that I hope are helpful to the authors.

1. The Introduction emphasizes endocasts repeatedly and the inability to examine the cerebellum properly in endocast analyses. This is true, but it has little to do with the paper presented. The authors are not doing an endocast analysis, they are using mCT to image the cerebellum itself in fluid preserved squamates. I would add that although digital endocasts are a hot and rapidly expanding topic at the moment, there is a long history of volumetric and stereological analyses of animal brains, including a classic paper by Platel on squamates that was oddly missing from the references section. The references to endocasts therefore need to be removed and more emphasis placed on brain evolution studies in general, especially those focused on the evolution of size and shape differences of brain regions.

2. Related to my previous point, several key references on cerebellar evolution were absent from the Introduction and the rest of the manuscript. For example, several key comparative studies on cerebellar evolution in birds and mammals published over the past 15 years were noticeably missing. More critically, two recent studies from the Wylie lab on lizard and rattlesnake cerebella, including a description of the disorganized Purkinje cells in the rattlesnake, were absent as was a reference to the lizard brain atlas recently published by Hoops et al. These studies seem crucial to the current manuscript for several reasons and need to be cited and discussed.

3. The bulk of the methods section was extremely difficult to read. A long list of R scripts is provided, but with little to no explanation regarding why these analyses are being done or what they do. For example, I am familiar with many phylogeny-informed statistical packages, but 'phylolm' is not one that is commonly used and has not been peer-reviewed, in contrast to others that would be suitable for this analysis, such as 'evomap'. The Johnson-Neyman procedure is also unusual and I am unaware of papers that have validated or assessed its use within a phylogenetic context. In fact, post-hoc comparisons continue to be an issue for most phylogeny-informed statistics. Related to this point, no information is provided on what phylogeny was used, how it was constructed or if phylogeny was incorporated into the geometric morphometric analyses. I was impressed that the authors went to the trouble of brain clearing and light sheet microscopy, but insufficient information was provided on quantification. A range of 250-750 Purkinje cells is very broad and without knowing how they were selected or quantified, it is not possible to interpret their results. Also lacking was an explanation of why the distance between Purkinje cells and the pial surface was measured. I can think of a couple of reasons, but none are provided. I also did not follow what analyses could be done of the species or locomotor groups when it sounded like $n = 1$ for each species.

4. I am not an expert in transcriptomics, but the inclusion of this data was justified insufficiently. What question is the transcriptomics approach trying to answer? Is it just whether there are differences among locomotor groups? If so, how does one know that these are differences due to locomotion and not phylogeny?

5. The classification of locomotor groups was another area that required further explanation and description. A list of the species and their locomotor classification is provided in the supplementary material, but these are not defined anywhere. Anytime you categorize a behaviour or other trait in a comparative study, it needs to be defined explicitly so that other authors can employ the same categorization (or not as the case may be) in future comparative studies. Without this information, it is not possible to determine what the actual behavioural differences are among the different groups.

6. The figures were all of high resolution, but I found them very difficult to follow. For Figures 2-4, the reader is expected to constantly refer back to Figure 1 to determine what each of the colours mean. This took me 3-4 reads to understand as it was not apparent in Figure 2, what the colours referred to and the Results section kept referring to Figure 2 to highlight differences between burrowers and other groups. Making it even more problematic was the inclusion of multiple colour schemes within these figures. For example, within Figure 2, 2c and 2e are using one rainbow of colours to refer to the locomotor groups and another to refer to "consensus shape changes" and relative cerebellar volume respectively. This is almost impossible to interpret, especially for readers with subtle colour blindness. Note that I put consensus shape changes in quotes because I could not determine from the Results section what this meant or what the consensus shape looked like. Interestingly, the figures in the supplemental material were far easier to follow and I strongly recommend that the authors make use of those and include them in the manuscript proper.

7. I would add that Figure 1 itself is also needlessly complicated. The snakes are monophyletic so I do not see a need to have them represented by a different symbol. A bracket labeled "snakes" would be sufficient. Also, the colour scheme is not conducive to being interpreted by readers with anomalous colour vision. I encourage the authors to seek out resources available online or consult Journal of Comparative Neurology guidelines on how best to use colour in figures so that readers with anomalous colour vision can still interpret the figures appropriately.

8. The Results section was unfortunately difficult to follow, partially due to the lack of clarity in the figures and methods. For example, on page 4, lines 3-4, the authors state "...lizards display an inverted tilting causing homologous dorsal regions to project towards opposite directions (Extended

Data Fig 1b, c).". I read this several times and I still cannot determine what the authors are trying to describe and examining the figure did not help. Similarly, the description of "a gradual morphological transition from snakes to quadrupedal lizards, passing through intermediate forms exhibited by limbless and limb-reduced lizards" is difficult to follow and not apparent in Figure 2. The placement of all of the statistical results in the Extended Data also did not help in reading through the Results as this forces the reader to flip back and forth. And the descriptions of cerebellar shape variation were extremely difficult to follow. Overall, the Results need better descriptions of the patterns observed, details of the statistics embedded within the text itself and more appropriate figures within the manuscript so that the differences among the locomotor modes are easier to see and interpret.

9. The morphometric analysis of overall brain shape seemed out of place. The authors introduce the cerebellum as the focus of the paper in the Introduction and should stick with the cerebellum. Bringing in overall brain shape confuses the reader and whether brain shape as a whole reflects anything about behaviour or the sizes of brain regions is unknown.

10. The authors state in other parts of the manuscript that analyzing the size of brain regions is insufficient for understanding brain evolution, yet cerebellar size is included in the current analysis. It might be more appropriate to tone down the criticism of volumetric studies and perhaps highlight more positively that size is only one metric and that other parameters can change independently of brain region volume (e.g., neuron sizes, neuron numbers, brain region shape).

11. In their interpretation of the results, the authors comment that squamates have an exceptional cerebellar diversity. I do not think this conclusion can be reached based on the data presented. Yes, squamates vary in cerebellar size, shape and perhaps Purkinje cell distribution, but not any more so than bony fishes or mammals. The key finding here, that locomotor mode is a strong predictor of cerebellar size, shape, Purkinje cell distribution and cerebellar transcriptome, is far more important. In fact, this suggests that a key change in behaviour and limb morphology is evolutionarily correlated with major changes in the cerebellum. Platel and other authors have hinted at something similar based on a more limited data set, but I think the authors sell themselves short in relating a major behavioural transition with major changes in a brain region.

Nature Communications manuscript NCOMMS-19-15106-T

Squamate brain evolution unveils multi-level adaptations of the vertebrate cerebellum

Simone Macrì, Yoland Savriama, Imran Khan & Nicolas Di-Poï

Point-by-point replies to the reviewers' and editorial comments and suggestions

We thank the Reviewers for the positive evaluation of our manuscript. We have carefully considered all the comments and recommendations and we explain below how we revised the entire paper to comply with most of these observations. We reproduce the referees' comments in italics and our responses and explanations are in plain text.

Reviewer #1 (Remarks to the Author):

In this study the authors performed comparative morphological analyses of extant reptilian brains with multiple experimental strategies. The authors particularly focused on the Squamate cerebellum and executed 3-D computed tomography, whole mount imaging of Purkinje cell distribution, and transcriptomics in adult cerebellum, which conducted substantial associations among specific characteristics in these traits and various locomotive behaviors in limbless/limb-reduced or quadrupedal characteristics independent to phylogenetic relationship. The authors concluded that existence of significant correlations among these traits and locomotive behaviors resulted from convergent adaptive evolution, and that mosaic patterns of brain evolution can be categorized by cerebrotypes based on their definition. Each part of analysis had been performed in detail with enough number of samples and data representation were convincing, however, there are conceptual problems in the integration of the results and ambiguity in the purpose of the project.

We thank this Reviewer for highlighting the appropriateness of our dataset and analyses as well as for her/his constructive comments that significantly improved our discussion on the functional importance of our data. However, the new experiments asked by this Reviewer are to our opinion beyond the scope of this study and beyond the scope of a revision, as they would require several years of specimen collection and experimental work (see below), and they are now mentioned in the revised manuscript as future project directions.

1) The most significant problem is on lack of the causality among morphological traits and specific locomotive behaviors across species. It is unsure how unique shape of the cerebellum is functionally associated with the pattern of locomotion, Purkinje cell distributions, and transcriptome data represented in the manuscript. Without experimental proves or any relevant explanations for the functional connections, all the represented results are still fragmental and hard to conduct the principle underpinning the brain evolution.

We agree that there are not yet any “direct” experimental links between most observed morphological traits and locomotor behaviour, but the main focus of our new study was to first identify any potential

relationships between cerebellar morphology and locomotor phenotype in a largely underexplored vertebrate group—squamates. Nevertheless, both qualitative descriptions over the last century and recent quantitative analyses in different vertebrate groups have highlighted correlations between cerebellar size and/or morphology and behavioural aspects involving organism interactions with the environment, among which locomotion is a predominant feature. For example, comparative bird studies focusing on various aspects of brain morphology have shown correlations between specific morphological traits and flight behaviour (Iwaniuk et al. *Brain. Behav. Evol.* 67:53, 2005; Iwaniuk et al. *Brain. Behav. Evol.* 68:45, 2006; Iwaniuk et al. *Brain. Behav. Evol.* 69:196, 2007), and some convergent pattern of cerebellar shape was even observed in bird species adopting similar prey and hunting behaviour. The vertebrate cerebellum is well-known to be divided into an elaborate array of parasagittal zones that form an exquisitely organized topographic map controlling sensory-motor behaviour (Apps & Hawkes *Nat. Rev. Neurosc.* 10:670, 2009), including in reptiles (Larsell *J. Comp. Neurol.* 41:59, 1926; ten Donkelaar & Bangma *Biology of Reptilia Vol. 17*, pp. 496–586, 1992). Larsell noted such morphological features of the squamate cerebellum a century ago (Larsell *J. Comp. Neurol.* 41:59, 1926), and he already proposed that the corpus cerebelli was divided into two topographic regions, the pars interposita (medial) and the pars lateralis (lateral), which are involved in the control of axial musculature and limb movement, respectively. Based on precise qualitative comparison of several species showing different degrees of limb reduction (*Sceloporus biseriatus*, *Gerrhonotus principis*, *Anniella nigra* and a snake, *Thamnophis sirtalis*), he concluded that the divergent development of these two cerebellar regions was correlated with both the degree of appendage development and method of locomotion. Importantly, similar differences in the size and degree of development of cerebellar regions were observed in our large dataset, and we further noticed a marked enlargement of the medial cerebellar area in our facultative bipedal/aerial quadrupedal lizards (i.e., *Basiliscus*), a group that requires trunk musculature support during execution of their sophisticated locomotor tasks. Furthermore, the presence/absence and degree of development of motor control brainstem nuclei and spinal pathways, including the red nucleus, vestibular nuclei, certain reticular nuclei, as well as the rubro-spinal, vestibulo-spinal, vestibulo-reticulo-spinal, and reticulo-spinal tracts, have been associated with the different body plans and locomotor behaviours of reptiles (Weston *J. Comp. Neurol.* 65:93, 1936; Stefanelli *Commentat. Pontif. Acad. Scient.* 8:147, 1944; Newman & Cruce *J. Morphol.* 173:325 1982; ten Donkelaar et al. *Anat. Embryol. (Berl.)* 168:277, 1983; ten Donkelaar *Behavi. Brain Res.* 28:9, 1988; ten Donkelaar & Bangma *Biology of Reptilia Vol. 17*, pp. 496–586, 1992), but also with their level of cerebellar complexity (Weston *J. Comp. Neurol.* 65:93, 1936; Stefanelli *Commentat. Pontif. Acad. Scient.* 8:147, 1944; ten Donkelaar *Behavi. Brain Res.* 28:9, 1988; ten Donkelaar & Bangma *Biology of Reptilia Vol. 17*, pp. 496–586, 1992). In our squamate dataset, the observed reorganization of specific cerebellar regions (in terms of size and shape), including between quadrupedal and limbless animals, is then highly expected to correlate with differences in the presence/absence and/or degree of development of brainstem nuclei and spinal pathways, particularly the red nucleus as well as the reticulo-spinal and rubro-spinal tracts (see below and reply to comment 2 of this Reviewer). For example, snakes but also limbless and limb-reduced lizards from our dataset show a reduction of the lateral extension of the cerebellum, which parallels previous observations related to a reduction in size of the red nucleus and absence of a rubro-spinal tract in boid snakes (ten Donkelaar *J. Comp. Neurol.* 167:421, 1976; ten Donkelaar *J. Comp. Neurol.* 167:443, 1976; ten Donkelaar *Brain Res.* 57:25, 1982; ten Donkelaar et al. *Anat. Embryol. (Berl.)* 168:277, 1983; ten Donkelaar & Bangma *Brain Res.* 279:229, 1983). Moreover, these animals display a relative enlargement of the medial cerebellar portion that parallels descriptions highlighting a large reticulo-spinal tract in *Python regius*

(ten Donkelaar et al. *Anat. Embryol. (Berl)*. 168:277, 1983). Altogether, such data definitely suggest that the observed re-organization in the shape, size and degree of development of cerebellar topographic regions are functionally associated with locomotor system anatomy (including limb morphology) and locomotor behaviours.

Regarding the Purkinje cell (PC) arrangement, numerous genetic and functional studies in both mice and organotypic cultures have demonstrated that an altered PC layout triggers both motor coordination and neuroanatomical defects (see, e.g., Falconer *J. Genet.* 50:192, 1951; Mariani et al. *Philos. Trans. R. Soc. Lond. B Biol. Sci.* 281:1, 1977; Goffinet *Int. J. Dev. Biol.* 36:101, 1992; Miyata et al. *J. Neurosci.* 17:3599, 1997). Furthermore, PC ectopically located in the molecular layer were found in several mouse models for syndromes causing motor dysfunctions such as ataxia-telangiectasia (A-T) and DYT1 dystonia (Borghesani et al. *Proc. Natl. Acad. Sci.* 97:3336, 2000; Song et al. *Neurobiol. Dis.* 62:372, 2014). Finally, the presence of ectopic PCs in the molecular layer of the cerebellum has been recently recognized as a clinical trait of several human locomotor disorders, including essential tremor (Kuo et al. *J. Neurol. Neurosurg. Psychiatry* 82:1038, 2011) and various types of spinocerebellar ataxias (Louis et al. *Cerebellum* 17:104, 2018), which cause kinetic tremor, gait ataxia and various levels of motor impairment. It would be of course very interesting to experimentally test the effect of snake PC positioning on locomotor behaviours, but it might be actually relatively complex as any attempt to alter the PC layout has so far been confronted to an impairment of the entire cerebellum development in mice.

As suggested by Reviewer 2, the functional relevance of genes identified in our transcriptomic study are now included and discussed in the revised “Results and Discussion” of the manuscript (including with new references on vertebrate gene functions). Most of the observed differentially expressed genes in the “locomotory behaviour” category are in fact already known to be predominantly expressed in the vertebrate brain and to play a key role in motor coordination, balance, and/or locomotor activity in multiple vertebrate models (see reply to Reviewer 2). These data highlight the relevance and importance of our transcriptome study for understanding the evolution of gene expression and the transcriptome-phenotype relationship (see also reply to comment 4 of this Reviewer).

Altogether, although we do not have yet any direct experimental links between cerebellar morphology and locomotor behaviour in squamates, we now provide relevant explanations for the functional connections of each observed morphological trait, including in the revised “Results and Discussion” section of the revised manuscript, by also including new references (see, e.g., new references 2,50-54,59-62,66,67). Furthermore, and also based on comment 4 of this Reviewer, we now highlight in the revised “Conclusions” section that our work provides a new framework for the evolution of cerebellar structure and locomotor behaviours in vertebrates, which can be reinforced by future experimental works and dissection of molecular and developmental mechanisms.

2) Several previous comparative studies of locomotion and relevant neural circuits revealed significant associations between the nucleus ruber and reticulo-spinal tracts and quadrupedal behaviors in reptiles, while the cerebellum is relatively undeveloped in these animal groups, leading the question why the authors focused on the cerebellum but not examined other nuclear structures functionally associated with locomotive behaviors.

Our study focus on the cerebellum simply because, although this structure does not initiate movement, it contributes to its coordination, precision, and accurate timing by receiving input from sensory systems of the spinal cord and from other parts of the brain, including the cerebral cortex, and by

integrating these inputs to fine-tune motor activity. Cerebellar circuits are thus of key importance in the control of movements, providing a neural basis for pattern recognition and motor behavioural correction and adaptation. Furthermore, the size and/or anatomical features of the cerebellum have been correlated for more than a century with various behavioural or ecological strategies in all vertebrate groups, including reptiles (see also reply to comment 1 of this Reviewer). As suggested by the Reviewer (and also discussed in the reply to comment 1 of this Reviewer), we agree that the investigation of major descending motor pathways such as the reticulo-spinal and rubro-spinal tracts could also lead to significant correlations with locomotor behaviours in squamates, as shown for example in mammals and fish. However, vertebrate control of locomotion involves multiple regulatory neuronal components and brain areas that work together to control its initiation, execution, and coordination, so investigation of the structures mentioned by this Reviewer would only be complementary to our data. Furthermore, phylogenetic comparisons of cerebellar circuits in vertebrates have highlighted key differences in both the degree of development and functional importance of some neuronal tracts in vertebrates (see, e.g., Ito *The Cerebellum and Neural Control* (New York: Raven Press), 1984). Especially, as highlighted above (see reply to comment 1 of this Reviewer), the absence of some pathways such as the rubro-spinal tract as well as the presence of a well-developed crossed rubro-bulbar projection in at least some snakes have suggested a predominant function of the red nucleus in the control of mastication rather than locomotion in this group (ten Donkelaar & Bangma *Brain Res.* 279:229, 1983). As a result, interspecific analysis of the red nucleus in the context of locomotor behaviour might be hindered by comparing structures that are devoted to different function in a dataset comprising quadrupedal and limbless animals. In contrast, though labelled as “underdeveloped”, the key role of the cerebellum in locomotor behaviour is established in reptiles and is based on its control over brain stem nuclei as well as direct spino-cerebellar pathways (Bangma et al. *J. Comp. Neurol.* 230:218, 1984; ten Donkelaar *Behavi. Brain Res.* 28:9, 1988; ten Donkelaar & Bangma *Biology of Reptilia Vol. 17*, pp. 496–586, 1992). In fact, the relatively simple morphology of the reptile cerebellum facilitated our study, as it allowed us to clearly underline differences otherwise difficult to highlight and quantify in more complex structures. We now revised the “Introduction” to better emphasize the critical role of the cerebellum in movement control and locomotion, and we also added the neural circuit aspect mentioned by this reviewer when referring to previously observed (in complement to anatomical features) adaptations to locomotor behaviour.

3) Adult brain morphology must be significantly affected by external physical constraint, i.e., the shape of skulls. For example, the length of olfactory tracts is tightly constricted by the morphology of the orbit in reptiles. The author have to elaborate how they could eliminate extrinsic influences to the brain shapes.

We agree that the pattern of skull ossification could potentially impose evolutionary constraints on cerebellar development, especially because cerebellar development in vertebrates (including in squamates) generally occurs during late embryogenesis and early postnatal period, but changes in the development, shape or structure of skull bones have been frequently reported to accommodate for brain morphology changes. For example, modification and/or absence of skull bone ossification and/or suture fusion/position may avoid physical constraints on brain morphology. In fact, the growth of the skull in harmony with the brain is an extremely complex morphogenetic process. Particularly, the multiple skeletal components of the skull originate asynchronously and their developmental schedule varies across vertebrates but also across squamates (see, e.g., our recent work Da Silva et al. *Nat. Commun.* 9:376, 2018), so the analysis of cerebellum-skull relationships would require a deep ontogenic investigation of morphological variations (using, e.g., geometric morphometrics) in both cerebellum

and braincase region of skull (including supraoccipital bone) from our multiple snake and lizard species. Although we agree that this brain-skull interaction aspect is extremely interesting, collections of lizard and snake specimens are sparse, so starting such work would already require a huge effort to obtain sufficient embryonic and postnatal series of our selected squamate species in good fixation/preservation conditions. Such analysis is definitively beyond the scope of a revision, as it could lead to an exciting separate publication in a high-impact journal (see, e.g., the recent publication by Fabbri et al. *Nat. Ecol. Evol.* 1:1543, 2017, focusing on the correlation between regions of the brain and bony skull element in Reptilia). Related to that, Reviewer 3 already suggested to “*consider breaking this manuscript up into more than one paper*”.

4) The project completely lacks the aspect of developmental biology that is critical to gain mechanistic insights into the evolution of species-specific body structures. “Developmental genes” in the adult tissue represented in the transcriptome data does not always reflect the functional association with embryogenesis, because many developmental genes are pleiotropic in their expressions and functions. The author must execute comparative ontogenic analysis of the cerebellum and cellular distributions in different species to clarify causal relationships among neuroanatomical architectures, gene expression, and unique locomotive behaviors.

Similarly to the previous comment (see comment 3 of this Reviewer), the investigation of the developmental mechanisms leading to all observed morphological traits (including through comparative ontogenic analysis, as proposed by the Reviewer) is beyond the scope of a revision. Indeed, this would first require the optimization of some genetic tools in different non-model squamate models, a huge effort to obtain fresh embryonic and postnatal series of various squamate species (see also reply to comment 3 of this Reviewer), and the precise developmental staging of all different selected models in order to compare equivalent developmental stages. Furthermore, the dissection of molecular and developmental mechanisms is beyond the scope of this “evolutionary biology” study, which rather focuses on identifying the significant relationships between several brain morphological traits and locomotor behaviours (see also title of the manuscript). Nevertheless, we now provide relevant explanations for the functional connections of each observed morphological trait with locomotor behaviour in the revised “Results and Discussion” section of the manuscript (see reply to comment 1 of this Reviewer). We fully agree that gene functions might change between embryonic and adult stages, but considering the timing of cerebellar morphogenesis that generally develops over a long time, extending from late embryonic period until the postnatal period, the evolutionary mechanisms that generate morphology and neural-circuit diversity of the cerebellum continue after birth (see, e.g., Hibi et al. *Dev. Growth Differ.* 59:228, 2017), including in squamates (personal observations). In fact, postnatal development of the cerebellum has been shown to be critical for its intact function such as motor coordination in mammals. During this period, the cerebellum undergoes dramatic changes as neural progenitor cells undergo mitosis, cell fate specification, radial migration, neurite growth, and spine formation. As now mentioned in the revised “Results and Discussion” section, the transcriptome analysis is then particularly relevant for cerebellar tissues that develop until the postnatal period (see other justifications of the transcriptomic approach in the reply to comment 4 of Reviewer 3). Furthermore, as already mentioned above (see reply to comment 1 of this Reviewer), most of the observed differentially expressed genes are in fact already known to be predominantly expressed in the vertebrate brain and to play a key role in motor coordination, balance, and/or locomotor activity in multiple vertebrate models (see reply to Reviewer 2), thus confirming the

relevance and significance of our conclusions on the transcriptome-phenotype relationship. The functional relevance of our transcriptomic study, including both in timing but also in gene expression pattern, is now included and discussed in the revised “Results and Discussion” of the manuscript (including with new references on vertebrate gene functions). Finally, and also based on comment 1 of this Reviewer, we now highlight in the revised “Conclusions” section that our work provides a new framework for the evolution of cerebellar structure and locomotor behaviours in vertebrates, which can be reinforced by future experimental works and dissection of molecular and developmental mechanisms.

Reviewer #2 (Remarks to the Author):

The authors present a unique and extended dataset of Squamata brains which allowed them to analyse the shape and size of the different regions. A clear correlation of the cerebellum shape is shown with the locomotion type of the species. The manuscript is well written and illustrated and the information is clear.

We thank this Reviewer for highlighting the clarity of our manuscript as well as for her/his constructive comments that significantly improved the whole manuscript, including the significance of our conclusions on the transcriptome-phenotype relationship. We agreed with most comments and suggestions and have revised the text and dataset information (see revised Supplementary Table 1) accordingly.

There are only a few points that I believe should be addressed:

- provide a table with the number of individuals analysed for each type of analyses (3D model, histology, light sheet, RNA seq) as well as their age/size. It is important to show and maybe indicate in the text that only adult-sized animals were included in the study

This is an understandable request because, as highlighted above by this Reviewer, the sampling of our “unique and extended dataset” has required a huge effort to obtain properly fixed or fresh samples from multiple non-model squamate species at corresponding age/size (see below). As already mentioned in the main text and now further emphasized in the “Methods” section of the revised manuscript, the number of analyzed individuals has obviously been limited by sampling difficulties for some particular lizard and snake species, including species with particular ecological behaviours (e.g., fossorial species living underground) and/or a geographically restricted area of distribution. As now mentioned in the “Methods” section of the revised manuscript, an average of 2.6 individuals per species was examined in our study (range 1-8 depending on sampling difficulties and type/number of analyses performed), and specimens were used for multiple observations and/or analyses whenever possible. Considering our main focus on comparing groups of individuals belonging to different species with specific locomotor behaviours (see also reply to comment 3 of Reviewer 3), but also the fact that all species were only used in 3D model analyses (only selected species were used in histology, light sheet, and transcriptome analyses), the number of individuals analyzed is higher than what is usually done in non-model squamate studies using interdisciplinary analyses on an extended dataset covering the entire phylogeny (see also general comment of Reviewer 1). To avoid the addition of a new Supplementary Table, as

proposed by this Reviewer (see remarks by Reviewer 3 on the Supplementary material), we rather incorporated both the total number (with types of analysis performed) and the size range of individuals used for each species in the revised Supplementary Table 1, and the number of individuals analyzed for each type of analysis is now given in the respective parts of the revised “Methods” section. We also now provide the age range of animals in the revised “Methods” section (young adult stage: 8-18 months after hatchling), but head length measurements was preferred over ages in the revised Supplementary Table 1 because of both major interspecies variations in average lifespan and intraspecies variations in postnatal growth rate in squamates (see our recent work: Eymann et al. *J. Comp. Neurol.* 527:2356, 2019).

- I am surprised by the small number of one-to-one orthologs, only 630 out of ~9000 unigenes. I believe that GO analyses are more meaningful when a larger set of genes is available, especially as one gene can belong to several categories. How many genes are shared by nine species and how different is the GO distribution of the genes in the three clusters?

We fully agree with this comment, and we suspect that the difficulty of comparing transcriptomes of non-model, distantly-related lizard and snake species with no or low-coverage genome data (9 out of 10 species analyzed) could largely explain the low number of orthologs shared across all ten species. Indeed, the use of transcriptomic data often prevents recovery of full-length transcripts, which could then hinder an accurate assessment of orthology by introducing a large amount of fragmentation, frameshifts, and/or mis-indexing in sequence analysis. In fact, a number of recent comparative studies from diverse animal groups have already reported a similar % of shared orthologs (< 10%) when using multiple species (see, e.g., Hawkins et al. *Proc. Natl. Acad. Sci USA* 116:11351, 2019; Shultz & Sackton *Elife* e41815, 2019 ; Fan et al. *PLoS One* 12:e0190023, 2017 ; Zhao et al. *Front. Plant Sci.* 7:732, 2016). Furthermore, previous studies have shown that ortholog identification varies greatly between species and gene classes (see, e.g., Derrien et al. *PLoS One* 7:e30377, 2012). Especially, squamates are a morphologically and ecologically diverse group characterized by a significant number of rapidly-evolving genes showing accumulation of amino acid changes (see our work: Di-Poi et al. *Nature* 464:99, 2010; Di-Poi et al. *Genome Res.* 19:602, 2009), including in key developmental and functional genes highly conserved across vertebrates, which are known to affect the sensitivity of ortholog identification (Martín-Durán et al. *Genome Res.* 27:1263, 2017). Finally, another contributing factor might be the presence of low expression values in at least one sample, as only orthologs expressed above a fixed expression threshold of 0.5 FPKM in at least one sample were retained in our study to exclude possible noise at very low expression levels and get a more accurate characterization and quantification of orthologous transcripts across species.

As suggested by the Reviewer, we also re-checked the number of orthologs shared by any combinations of nine species, and the % still remains relatively similar (7% of orthologs recovered using 10 species, and range 7.4-9% using only 9 species), indicating that none of the species/transcriptomic data are problematic in our study. Furthermore, as already mentioned in the main text, housekeeping and structural genes such as regulators of cytoskeletal actin show relatively similar expression values across species, thus confirming the validity of our data and experimental design. Similarly, the GO distribution is very similar in the three different clusters for high-level biological process terms, with more than 50% of genes belonging to the same top categories at both levels 2 (e.g., cellular process, biological regulation, response to stimulus, developmental process) and 3 (e.g., regulation of cellular process, cellular component organization, cellular response to stimulus).

We now better detailed and justified these two key points on the number of one-to-one orthologs and GO terms in the “Results and Discussion” and “Methods” sections of the revised manuscript, by also including new references.

- The brain is known to express a large number of genes where their function is often unknown. In cluster 3, the authors detect 12% of the genes (how many genes does this correspond to?) linked to locomotory behaviour. I believe it is important to discuss further these findings. Which genes are these? Why would their expression in the brain affect the locomotion of the animal? How relevant or random is this finding, compared for example to the 'response to inorganic substance' or 'multicellular organism reproduction' in clusters 1/2?

We agree with this comment and have re-checked carefully our gene lists. The number, names, and relevance of genes identified in the “locomotory behaviour” are now included and discussed in the revised “Results and Discussion” of the manuscript (including with new references on vertebrate gene functions). The percentage indicated in the new Figure 5b for “locomotory behaviour” corresponds to 29 genes, thus representing a good proportion (15%) of the total number of genes present in this category on the Gene Ontology Resource, considering that this database includes genes expressed in any tissues besides the cerebellum/brain. Importantly, the “locomotory behaviour” category is defined as a specific movement from place to place of an organism, and most of the observed differentially expressed genes in our study are in fact already known to be predominantly expressed in the vertebrate brain and to play a key role in motor coordination, balance, and/or locomotor activity in multiple vertebrate models such as zebrafish (e.g., Spatacsin), mice (e.g., Contactin-2, Astrotactin-1, Ephrin type-A receptor 4, striatin, Alsin, Metabotropic glutamate receptors 1 and 5, Nuclear receptor coactivator 2, Hamartin), rats (e.g., Unconventional myosin-Va), and humans (e.g., Pumilio homolog 1). Here are a two detailed examples that are now mentioned in the revised manuscript: knock-out mice for Contactin-2, a cell adhesion molecule expressed in the Purkinje fiber network and critical for neuronal patterning and ion channel clustering, exhibit severe ataxic phenotype consistent with defects in the cerebellum (Berglund et al. *Neuron* 24:739, 1999); mice that lack Astrotactin-1 have abnormal development of Purkinje cells associated with defects in balance and coordination as well as walking behaviour (Adams et al. *Development* 129:965, 2002). The two other categories mentioned by this Reviewer (“response to inorganic substance” and “multicellular organism reproduction”) are much broader categories on the Gene Ontology Resource (>2000 genes in total), and do not include any of the genes identified in the “locomotory behaviour” category. In fact, their definitions are not directly or exclusively linked to the locomotion behaviour of an organism, as “multicellular organism reproduction” refers to a biological process in which new individuals are produced by one or two multicellular organisms, and “response to inorganic substance” refers to a process that results in a change in state or activity at any levels (including molecular or cellular levels), thus supporting the similar expression pattern observed across species. We really thank the Reviewer for this suggestion, which further highlight the relevance and importance of our transcriptome study for understanding the evolution of gene expression and the transcriptome-phenotype relationship.

Reviewer #3 (Remarks to the Author):

The manuscript submitted by Macri et al. is a very ambitious, inter-disciplinary approach to examining brain evolution in a clade that has been the focus of relatively few studies: squamates. An impressive suite of approaches were brought to bear on cerebellar anatomy specifically and there are some interesting and very novel findings presented.

We thank this expert for highlighting the importance of our new “impressive” inter-disciplinary study in helping to clarify brain evolution in a largely underexplored vertebrate group.

However, the results are presented in a manner that I found extremely confusing and it was difficult to distill what conclusions could be made. The over emphasis on referring to the supplemental material detracted from the manuscript itself and the figures presented within the manuscript were difficult to understand. Adding to the confusion and interpretability was a densely written methods section that did not adequately explain why certain analyses and tests were done and introduced others that I do not think are appropriate in the context of phylogeny-informed analyses. I strongly recommend that the authors consider breaking this manuscript up into more than one paper or significantly expanding the current one so that the reader can understand what was done and why. I provide more detailed comments below that I hope are helpful to the authors.

We really thank this Reviewer for her/his constructive comments that significantly improved the whole manuscript. We strongly suspect that the lack of manuscript clarity mentioned by this Reviewer (although contradicting with comments from other Reviewers) was due to the condensed format of the manuscript initially formatted for Nature. We agreed with most comments and suggestions and have substantially revised the text and Figures, in many respects. Particularly, Figures have been simplified, improved, and/or split (see new Figures 2 and 3 that replace Figure 2 of initial submission), all sections of the text have been expanded and clarified, and key data included in the Supplementary information have been embedded within the main text. We provide below the replies to all her/his comments.

1. The Introduction emphasizes endocasts repeatedly and the inability to examine the cerebellum properly in endocast analyses. This is true, but it has little to do with the paper presented. The authors are not doing an endocast analysis, they are using mCT to image the cerebellum itself in fluid preserved squamates. I would add that although digital endocasts are a hot and rapidly expanding topic at the moment, there is a long history of volumetric and stereological analyses of animal brains, including a classic paper by Platel on squamates that was oddly missing from the references section. The references to endocasts therefore need to be removed and more emphasis placed on brain evolution studies in general, especially those focused on the evolution of size and shape differences of brain regions.

We only partially agree with this comment as both whole-brain and endocast approaches have been important in the field to assess the ecological significance of brain size and morphology in multiple vertebrate groups, and a recent publication has demonstrated the overall strong size and shape correlations between most brain subdivisions and endocasts (Watanabe et al. *J. Anat.* 234:291, 2019). However, as already mentioned in the “Introduction” section, we agree that volumetric and stereological analyses of animal brains have been more numerous and conclusive so far in the context of cerebellar structure, partially because of the poor shape correspondences of endocasts in the hindbrain region. We now better emphasized the volumetric and stereological studies in the “Introduction” section of the revised manuscript (also based on comment 10 of this Reviewer), by also adding additional comparative brain studies published in several vertebrate groups (see comment 2 of

this Reviewer). As suggested by the Reviewer, we also removed most references to endocasts in the manuscript, including in the revised “Abstract”, but we still kept a short mention of endocast studies in the revised “Introduction”, as the low hindbrain-endocast relationships also justify our use of alternative 3D high-resolution approaches. We also included the reference mentioned by this Reviewer (Platel *J. Hirnforsch.* 17:513, 1976) in the revised “Introduction”, when referring to previously observed brain-ecological relationships (although no formal analysis was performed in this publication) and morphological diversity of squamate cerebellum.

2. Related to my previous point, several key references on cerebellar evolution were absent from the Introduction and the rest of the manuscript. For example, several key comparative studies on cerebellar evolution in birds and mammals published over the past 15 years were noticeably missing. More critically, two recent studies from the Wylie lab on lizard and rattlesnake cerebella, including a description of the disorganized Purkinje cells in the rattlesnake, were absent as was a reference to the lizard brain atlas recently published by Hoops et al. These studies seem crucial to the current manuscript for several reasons and need to be cited and discussed.

We now carefully re-checked the cerebellum literature, and 14 new key publications on cerebellar evolution in multiple vertebrate groups, including birds, mammals, anurans, and squamate reptiles (including reference mentioned in comment 1 of this Reviewer) were added to the revised manuscript (see new references 3,4,6-8,14,16,23-26,30,31,33). Furthermore, 4 new references were added on mosaic evolution of brain structure in mammals and birds (see new references 70-73), as well as 8 references on the general organization of the squamate cerebellum (see new references 37-39 mentioned by this Reviewer) and motor control pathways (see new references 50-54, based on comment 1 of Reviewer 1). All new publications are now cited and discussed in several parts of the revised manuscript. For example, the squamate references mentioned by the Reviewer are now cited in the revised “Introduction” to better highlight the morphological diversity of squamate cerebella, thus further justifying our choice of the squamate model in this study. The same publications are also now mentioned in the revised “Results and Discussion” section when comparing the Purkinje cell phenotype across squamates, as similar differences in lizard and snake PC patterns were already noticed in these publications based on dragon lizard and western diamondback rattlesnake data.

3. The bulk of the methods section was extremely difficult to read. A long list of R scripts is provided, but with little to no explanation regarding why these analyses are being done or what they do. For example, I am familiar with many phylogeny-informed statistical packages, but ‘phylolm’ is not one that is commonly used and has not been peer-reviewed, in contrast to others that would be suitable for this analysis, such as ‘evomap’.

A number of different functions and R scripts were used to run morphometric analyses but also to make high-quality Figures (as highlighted by this Reviewer in point 6 below). We agree that the list of script is relatively long, and we now simplified this methods section by only keeping packages used to run the different geometric morphometric and statistical analyses. The use/function of most packages was in fact already justified in the initial manuscript (with website link and citation), but we now provide additional details for some analysis/package (including new citations on ‘phylom’ package and phylogeny, see comments below). The ‘phylom’ package has been published in 2014 in the peer-reviewed scientific journal “Systematic Biology” (see Tung Ho & Ane *Syst. Biol.* 63:397, 2014), and it has already over 200 citations, including in some studies on lizard brains (see, e.g., Hoops et al. *Brain*

Behav. Evol. 90:211, 2017). The reference for this package was missing in the initial submission, but it is now mentioned in the revised “Methods” section.

The Johnson-Neyman procedure is also unusual and I am unaware of papers that have validated or assessed its use within a phylogenetic context. In fact, post-hoc comparisons continue to be an issue for most phylogeny-informed statistics. Related to this point, no information is provided on what phylogeny was used, how it was constructed or if phylogeny was incorporated into the geometric morphometric analyses.

As already mentioned in the “Results and Discussion” and “Methods” of the manuscript, intergroup comparisons were initially made using phylogenetic ANCOVA analysis. However, a requirement of ANCOVA is that the relationship with the covariate is uniform across groups, meaning that regression slopes are homogenous. When the scaling exponents differ between groups, it is not possible to compare them using ANCOVA. The alternative Johnson-Neyman technique was then used to identify the range of values for which two groups are not significantly different. This technique has previously been applied to the fields of medical and behavioural science, sociology, ecology and comparative physiology, including for the analysis of animal growth and development, metabolic physiology, and digestive physiology (see many publication examples in White & Kearney *Compr. Physiol.* 4:231, 2014). The major assumptions of the Johnson-Neyman technique are similar to those of ANCOVA, but unfortunately there is not yet a phylogenetically informed implementation of the Johnson-Neyman technique. Nevertheless, the technique has also been applied in studies that incorporate phylogenetic information (e.g., Lavin et al. *Physiol. Biochem. Zool.* 81:526, 2008; White *Physiol. Biochem. Zool.* 76:122, 2003), by using a low significance level to compensate for the lack of phylogenetic information incorporated in the Johnson-Neyman technique; this was considered appropriate because phylogenetically informed statistical methods typically have confidence intervals wider than those calculated using conventional statistical methods (Garland et al. *Am. Zool.* 39:374, 1999). In our study, the phylogenetic signal obtained with ANCOVA analysis was negligible (equivalent to zero), so the Johnson-Neyman method could be confidently used. We now better justified the choice of this method in the revised “Results and discussion” and “Methods” sections of the manuscript. Furthermore, as already mentioned in the manuscript, we used an alternative phylogenetic ANOVA analysis to confirm the significance of size variations among locomotor groups. Concerning the phylogeny, we used the most inclusive and recent phylogenetic studies available for extant squamate species (Tonini et al. *Biol. Conserv.* 204:23, 2016). This phylogeny contains all species analyzed in our study, and we now mentioned it in the revised “Methods” section (in addition to its initial mention in the legend of Figure 1). All morphometric and statistical analyses have been indeed corrected for phylogeny (see, e.g., “phylogenetic” ANOVA and “phylogenetic” ANCOVA mentioned in the text), except for the Johnson-Neyman method (see above) that lacks implementation of phylogenetic information (see justification of the method above and in the revised manuscript). This key information is now more apparent in the revised manuscript.

I was impressed that the authors went to the trouble of brain clearing and light sheet microscopy, but insufficient information was provided on quantification. A range of 250-750 Purkinje cells is very broad and without knowing how they were selected or quantified, it is not possible to interpret their results. Also lacking was an explanation of why the distance between Purkinje cells and the pial surface was measured. I can think of a couple of reasons, but none are provided. I also did not follow

what analyses could be done of the species or locomotor groups when it sounded like $n = 1$ for each species.

Cells were identified based on immunostainings against Calbindin-1, a specific marker of Purkinje cells, and all cells (i.e., no cell selection and/or exclusion) from similar cerebellar areas were quantified across species, thus explaining the broad range of cells depending of species-specific cerebellum size. Due to high intra- and interspecies heterogeneity in molecular layer (ML) thickness that could affect the relative positioning of PCs, particularly in species with scattered PC organization, individual PC distances were all normalized to the whole ML thickness by measuring the distance between the inner border of granule cell layer and outer border of ML (pial surface) at each specific PC location. As shown by the multiple values per species in new Figure 4c, the cell position distributions were in fact compared, so the analysis was performed using $n = 250-750$ observations (i.e., individual PCs) per species. Furthermore, the main goal of this study was to compare different locomotor groups, so individuals belonging to different species with similar locomotor behaviours were treated together as biological replicates. However, a few locomotor groups contain observations from only one species, depending on sample availability (as already mentioned and now further highlighted in the manuscript, see also reply to Reviewer 2), likely explaining the lack of significant segregation between some locomotor groups in our analyses. Nevertheless, the low number of species for some locomotor modes does not affect the overall significance of our data, as significantly different PC organization was also confirmed by directly comparing the obtained 4 major groups (groups I-IV, containing 3 or 4 species per group in this case) with similar statistical analysis (p -values < 0.0001). We now clarified and provided more details on the cell identification, quantification, and statistical analyses in the revised “Results and Discussion”, “Methods”, and “Figure 4 legend” sections.

4. I am not an expert in transcriptomics, but the inclusion of this data was justified insufficiently. What question is the transcriptomics approach trying to answer? Is it just whether there are differences among locomotor groups? If so, how does one know that these are differences due to locomotion and not phylogeny?

Differences in gene regulation and expression pattern have long been recognized as crucial contributors to the phenotypic diversity of nervous system development and function. The accurate characterization and quantification of orthologous transcripts across species are especially critical for understanding the evolution of gene expression and the transcriptome-phenotype relationship. Previous comparative studies have shown that evolutionary changes in gene expression contribute to behavioural phenotype and play a key role in phenotypic changes between species. Here, we employed comparative transcriptomics to determine the similarity of cerebellar gene expression across representative squamate species with similar or different locomotor behaviours. In addition to providing a molecular read-out of developmental processes and/or mechanisms underlying tissue diversification, the other goal of the transcriptomic approach was to specifically test whether differences in the species expression profiles are shaped more by their phylogenetic relationships or by differences in locomotor behaviours. For this purpose, we performed heat map analysis and hierarchical clustering to build a tree-like structure (dendrogram) where samples are clustered into hierarchies according to their degree of expression pattern similarity. As seen in the new Figure 5c, our expression phylogeny from orthologous transcripts clearly differs from the molecular phylogeny expected for squamates. Furthermore, the majority of the groupings in the expression phylogeny have relatively high bootstrap

support, thus strongly supporting that lizard and snake species cluster according to locomotor modes rather than phylogenetic relationships. We now clarified and provided more details on the goals and output of the transcriptome analysis in the revised “Results and Discussion” section of the manuscript, also based on comments of Reviewer 2.

5. The classification of locomotor groups was another area that required further explanation and description. A list of the species and their locomotor classification is provided in the supplementary material, but these are not defined anywhere. Anytime you categorize a behaviour or other trait in a comparative study, it needs to be defined explicitly so that other authors can employ the same categorization (or not as the case may be) in future comparative studies. Without this information, it is not possible to determine what the actual behavioural differences are among the different groups.

We totally agree with this comment and we now provide all definitions of categorization in the “Methods” section of the revised manuscript. Briefly, as described in the Supplementary Table 1, locomotor groups were defined by cross-correlating data from reptile databases, available literature, and personal observations on different criteria: cranial and post-cranial anatomical features, habitat modes, and movement types associated with locomotor performance and species-specific locomotor typologies. Most of our categorizations for individual criteria are following previous publications and definitions (see Supplementary Table 1 and new citations in revised “Methods”).

6. The figures were all of high resolution, but I found them very difficult to follow. For Figures 2-4, the reader is expected to constantly refer back to Figure 1 to determine what each of the colours mean. This took me 3-4 reads to understand as it was not apparent in Figure 2, what the colours referred to and the Results section kept referring to Figure 2 to highlight differences between burrowers and other groups. Making it even more problematic was the inclusion of multiple colour schemes within these figures. For example, within Figure 2, 2c and 2e are using one rainbow of colours to refer to the locomotor groups and another to refer to “consensus shape changes” and relative cerebellar volume respectively. This is almost impossible to interpret, especially for readers with subtle colour blindness. Note that I put consensus shape changes in quotes because I could not determine from the Results section what this meant or what the consensus shape looked like. Interestingly, the figures in the supplemental material were far easier to follow and I strongly recommend that the authors make use of those and include them in the manuscript proper.

As highlighted by the Reviewer, the main Figures were produced at high-resolution and should be kept in the main manuscript. However, all Figures have been now simplified, improved, and/or split (see below). For example, all Figures were revised to include “their own” colour-code legend. We also simplified and splitted Figure 2 into two new Figures, in order to only keep the cerebellum shape data in Figure 2 (according to comment 9 of this Reviewer below); this allowed us to add extra representative cerebella in the morphospace (see Revised Figure 2a) but also to increase the overall size of the two panels on cerebellar morphology (see Revised Figure 2a, b), thus improving our general descriptions of cerebellar diversity (based on comment 8 of this Reviewer). The volumetric data are now displayed in a new Figure 3, then heatmaps with similar colour-code (as previously shown in Figures 2c and 2e) are shown in separate Figures. Furthermore, to only use one colour-code, coloured symbols corresponding to the different locomotor modes were removed from revised Figures 2b and 3b, and only the names of the different locomotor groups are now shown. The combination of colours used in our initial Figures were already adapted for many different kinds of colour blindness, but we

still improved (and tested using colourblind simulators) our colour palettes in accordance with *Journal of Comparative Neurology guidelines* (see comment 7 of this Reviewer). We also modified the opacity of 3D ellipses in PCA plots (Figure 2a), confidence intervals (Figure 3a), as well as “violins” and “cells” in PC violin plots (Figure 4c) to improve the overall differences between colours. Finally, the “consensus” term was referring to the “overall mean shape in multidimensional space”, and we now replaced this term by its true definition at different positions within the revised manuscript.

7. I would add that Figure 1 itself is also needlessly complicated. The snakes are monophyletic so I do not see a need to have them represented by a different symbol. A bracket labeled “snakes” would be sufficient. Also, the colour scheme is not conducive to being interpreted by readers with anomalous colour vision. I encourage the authors to seek out resources available online or consult Journal of Comparative Neurology guidelines on how best to use colour in figures so that readers with anomalous colour vision can still interpret the figures appropriately.

The use of different symbols is critical for identifying the respective positions of individual lizard and snake species in Figures 2-4, especially for convergent lizard and snake species with similar locomotor behaviours, so we used a similar legend and colour-code throughout the whole manuscript (including in Figure 1) to help the readers. Alternate options, including colour-coded species names and/or colour-coded branches of phylogenetic tree, do not improve the quality and understanding of this Figure. In addition, a similar code for lizards and snakes has already been used in many publications, including from our laboratory (see, e.g., Da Silva et al. *Nat. Commun.* 9:376, 2018). However, to help the readers, we now improved this Figure by expanding the size of the different colour-coded symbols and by adding two brackets “snakes” and “lizards” in the phylogeny, as proposed by the Reviewer for snakes. Furthermore, our colour palettes have been revised and are now in accordance with *Journal of Comparative Neurology guidelines* (see also comment 6 of this Reviewer). Finally, most colour-coded symbols and panels from main Figures have been also revised and increased in size, and colour opacity has been improved in Figures 2a, 3a, and 4c (see also reply to comment 6 of this Reviewer).

8. The Results section was unfortunately difficult to follow, partially due to the lack of clarity in the figures and methods. For example, on page 4, lines 3-4, the authors state “...lizards display an inverted tilting causing homologous dorsal regions to project towards opposite directions (Extended Data Fig 1b, c).”. I read this several times and I still cannot determine what the authors are trying to describe and examining the figure did not help. Similarly, the description of “a gradual morphological transition from snakes to quadrupedal lizards, passing through intermediate forms exhibited by limbless and limb-reduced lizards” is difficult to follow and not apparent in Figure 2. The placement of all of the statistical results in the Extended Data also did not help in reading through the Results as this forces the reader to flip back and forth. And the descriptions of cerebellar shape variation were extremely difficult to follow. Overall, the Results need better descriptions of the patterns observed, details of the statistics embedded within the text itself and more appropriate figures within the manuscript so that the differences among the locomotor modes are easier to see and interpret.

All Figures and Methods have been now improved and clarified in the revised manuscript (see replies to comments 3-7 of this Reviewer). Furthermore, we now provide better descriptions of both whole-brain and cerebellum pattern changes observed in the PCA plots (new Figures 2a and new Supplementary Figure 3), and all sentences mentioned by this Reviewer have been revised. To help the

reader, additional representative 3D models of whole-brain and cerebellum (with species names; see new Figures 2a and new Supplementary Figure 3) were added to the PCA plots, and our new shape descriptions in the revised “Results and Discussion” section now include examples of species present in the PCA plots. Relative to the “tilting”, extra-information, including organ orientation and brain area names, were further added to sagittal sections shown in Supplementary Figure 1 to better highlight the divergent spatial orientation relative to the brain axes displayed by the cerebellum of lizards and snakes. PCA plots and 3D models (including in Figures 2a, b and new Supplementary Figure 3) were also increased in size for better visualization of shape variations. Concerning the statistics, key values from all tables shown in Supplementary Information are now embedded with the text, as suggested by this Reviewer.

9. The morphometric analysis of overall brain shape seemed out of place. The authors introduce the cerebellum as the focus of the paper in the Introduction and should stick with the cerebellum. Bringing in overall brain shape confuses the reader and whether brain shape as a whole reflects anything about behaviour or the sizes of brain regions is unknown.

We totally agree with this comment, as no significant correlation was in fact observed between whole-brain and locomotion behaviour, so the whole-brain shape data is now shown in Supplementary Information (new Supplementary Figure 3). For a better understanding of shape variations, additional 3D models and species examples were still added to the PCA plot (see reply to comment 8 of this Reviewer).

10. The authors state in other parts of the manuscript that analyzing the size of brain regions is insufficient for understanding brain evolution, yet cerebellar size is included in the current analysis. It might be more appropriate to tone down the criticism of volumetric studies and perhaps highlight more positively that size is only one metric and that other parameters can change independently of brain region volume (e.g., neuron sizes, neuron numbers, brain region shape).

We agree with this comment, and we now revised the “Introduction” and “Results and Discussion” sections of the manuscript to tone down our criticism about the brain volumetric analyses. For example, as suggested by this Reviewer, we now mention that “although brain size has been traditionally the preferred measured trait in past evolutionary studies, we show here that size is only one metric, and other parameters can change independently of brain region volume”. New references were also added to the revised “Introduction” (based on comment 2 of this Reviewer) to better highlight previous works published on other size-independent parameters such as morphological complexity and cellular content.

11. In their interpretation of the results, the authors comment that squamates have an exceptional cerebellar diversity. I do not think this conclusion can be reached based on the data presented. Yes, squamates vary in cerebellar size, shape and perhaps Purkinje cell distribution, but not any more so than bony fishes or mammals. The key finding here, that locomotor mode is a strong predictor of cerebellar size, shape, Purkinje cell distribution and cerebellar transcriptome, is far more important. In fact, this suggests that a key change in behaviour and limb morphology is evolutionarily correlated with major changes in the cerebellum. Platel and other authors have hinted at something similar based on a more limited data set, but I think the authors sell themselves short in relating a major behavioural transition with major changes in a brain region.

We agree with this comment, and both our “Abstract” and “Conclusions” sections were revised to tone down the “exceptional” cerebellar diversity and better highlight the unique relationships between ecological behaviour and cerebellum specialization.

Reviewers' Comments:

Reviewer #2:

Remarks to the Author:

The authors have fully addressed my comments and modified the manuscript accordingly.

Reviewer #3:

Remarks to the Author:

Although Macri et al. have undertaken significant revisions to their manuscript, it still lacks focus and some the claims made are not necessarily supported by their analyses or are confirmations of results published elsewhere. In particular, the repeated reference to the "accuracy" of their brain reconstructions and analyses is an over-statement and I have some concerns over tissue quality based on the sources of some of their material.

1. The Introduction still lacks focus and has some inaccuracies. For example, the first paragraph is far too general and the statement about studies not describing brain shape or internal neuroanatomy is a broad generalization and largely untrue. There many papers that have described brain shape accurately and internal anatomy across species. I think the point the authors are trying to make is that there few, if any, studies that have integrated multiple approaches to study brain evolution within a clade. That statement is entirely correct and what the authors have attempted to execute here.

The choice of cerebellum as a brain region to focus on is not sufficiently justified. There is no evidence that is "the most powerful model system for studying brain evolution". In fact, there is still relatively little comparative research on cerebellar anatomy, despite the detailed descriptions of Larsell, who first revealed the neuroanatomical diversity of the cerebellum across vertebrates. Further, whether the cerebellum can be identified on an endocast or not is irrelevant to the study of cerebellar evolution across extant species.

Several other statements were unclear or vague, such as "cellular content", "multi-level, interspecific variations" and the entire sentence on page 3 (lines 10-14). I think what they are trying to say is that there is a lot interspecific variation, but the extent to which this is reflected in numbers or sizes of neurons, morphology, behaviour or ecology is unclear. This is not, however, how this comes across in this paragraph.

In the last paragraph of the Introduction, I will first note that the cerebellum is not a "trait" in this context. The authors also do not explain what is meant by "multi-level specializations" and the entire sentence does not read like a hypothesis. This paragraph also lacks any predictions or a sense of what will be specifically tested in this study, including the methods to be used.

2. One of my main concerns with the analyses presented is the source of the brain material. First, no information is provided on fixation and different fixatives can have different effects on brain morphology and the amount of shrinkage relative to fresh brains. Second, museum collections typically fix specimens in formalin, but then store them long term in 70% ethanol. If the museum specimens were stored in ethanol, this can cause major changes in brain morphology and brain region sizes as a result of much greater dehydration than keeping brains in formalin. If this was the case here, the museum specimens could be biasing the analyses or providing inaccurate reconstructions of brain and cerebellar shape. Third, using a variety of different fixatives and storage procedures could affect the analyses across species, especially if the majority of a locomotor-group or clade came from a single source. The Supplementary Material lacks information on sources of the material, fixatives

and other details, so it was not possible to evaluate if these potential confounds were affecting their data. Fourth, using contrast enhancers can also introduce shrinkage and other artifacts that are not discussed and accounted for. If this is layered on top of the fixation/preservation issues I mention above, then there might be a range of artifacts affecting cerebellar morphology that could impact the results. These issues need to be addressed in order to have confidence in the results as presented.

3. Throughout the Results, there is a heavy emphasis on the Supplementary Material. Although I understand that journals such as this have strict page limits, the 20+ references to the Supplementary Material indicate that Supplementary Figures 1, 3 and 4 need to actually be embedded in the text of the manuscript proper.

There are also some inaccuracies in the text of the Results. For example, the authors claims that they "quantified the accurate shape of the brain" for the first time in a squamate. This is not true; Hoops et al. (2018) published an MRI brain atlas of a lizard (<https://doi.org/10.1002/cne.24480>) that is far more detailed in its anatomical description than what is provided here. The reference to endocast analysis is also irrelevant here.

I will add that the first part of the Results section was extremely difficult to read because it is presented as a single paragraph 3+ pages long with multiple reference to the Supplementary Material.

4. One page 7, line 9 there is a reference to "data not shown". This needs to be shown to demonstrate that this statistical assumption was violated and it could easily be included in the Supplementary Material. On the same page, line 23 there is a reference to "other parameters". This needs to be spelled out as there are some variables examined in this manuscript and many others that are not (e.g., neuron numbers, neuron sizes, neuronal complexity, neuropil volume).

5. In the analysis of Purkinje cell layout, some of the findings were over-stated. For example, the "remarkable" variation in Purkinje cell layer organization across squamates has been described before. The authors acknowledge that part-way through this section, but the description of this variation needs to be tempered to reflect the previous observations. I was also confused by what was means by "directly comparing the obtained four groups with similar statistical analysis". Last, the link between locomotor disorders and squamate locomotion seems a bit of a stretch. Yes, some locomotor disorders are associated with ectopic Purkinje cells, but this makes it sound as if the evolution of burrowing/lateral undulating is the result of cerebellar disorganization when that is unlikely the case. The evolution of different locomotor strategies might require different kinds of cerebellar mediated coordination, resulting in a different topological organization of the Purkinje cells, which is different to abnormal cerebellar development in mutant mice or humans with neurological disorders.

6. On page 9, lines 4-7, I could not determine what the authors were trying to say here and "personal observations" are not sufficient for a major journal article to support this statement.

7. Although the gene expression part of this paper is better described in this version, lines 9-12 on page 9 are better suited to the Introduction, which lacks any description or mention of transcriptomics. I would also say again, that linking some of the findings to mutant mice is a stretch and implies here that limbless species are uncoordinated, when they are not. Limbless species have a different type of coordination than quadrupedal squamates because the locomotor pattern is different.

8. Like the Introduction, the Conclusions lacks focus and contains inaccurate and/or vague phrases. For example, there is no means of determining whether the current analyses are more or less accurate than any others. Further, the analyses presented are not as comprehensive as claimed. Currently, most studies of this nature quantify neuron numbers and there is a profound lack of

information on variation in neuron numbers and neuronal morphology across species, for almost every vertebrate clade. This is not acknowledged anywhere in the manuscript, but should be discussed here and elsewhere. I will also add that the current study does not actually reflect “cerebrotype” analyses, which would be dependent on analyses across multiple brain regions or, at the very least, quantification within the cerebellum.

9. The methods describes manual segmentation about several brain regions, but there is no actual analyses of these presented. Morphometrics can be done without segmenting individual brain regions, so an explanation is needed for segmenting non-cerebellar brain regions and an analysis provided (which would be a cerebrotype-analysis), or some of this methods section removed.

10. Minor issues.

- a. There were at least two instances of “However, although”, which is grammatically incorrect.
- b. “neuron organization” is not a term
- c. Page 4, lines 29-30. The sentence that begins on line 29 needs to be rephrased.

Point-by-point replies to the reviewers' and editorial comments and suggestions

We thank again the Reviewers for the positive evaluation of our manuscript. We have carefully considered all the new comments and recommendations raised by Reviewer 3 and we explain below how we revised the entire paper to comply with most of these observations. We reproduce the referees' comments in italics and our responses and explanations are in plain text.

Reviewer #3 (Remarks to the Author):

Although Macri et al. have undertaken significant revisions to their manuscript, it still lacks focus and some the claims made are not necessarily supported by their analyses or are confirmations of results published elsewhere. In particular, the repeated reference to the “accuracy” of their brain reconstructions and analyses is an over-statement and I have some concerns over tissue quality based on the sources of some of their material.

1. The Introduction still lacks focus and has some inaccuracies. For example, the first paragraph is far too general and the statement about studies not describing brain shape or internal neuroanatomy is a broad generalization and largely untrue. There many papers that have described brain shape accurately and internal anatomy across species. I think the point the authors are trying to make is that there few, if any, studies that have integrated multiple approaches to study brain evolution within a clade. That statement is entirely correct and what the authors have attempted to execute here.

The first paragraph is general on purpose, as details on previous studies (including with references) are given in the second paragraph (see also additional details now given in our revised “Introduction” section). This paragraph briefly summarizes previous studies/works that have been done in an environmental or ecological context, and as also mentioned later by this reviewer (see comment 3), publications such as Hoops et al. 2018 show “detailed anatomical description” of a lizard brain which, although very useful and nicely illustrated, do not show any quantitative “shape” data, geometric morphometric analysis, or correlation with ecological parameters (see below and reply to comment 3). By “integrating multiple approaches” (as highlighted by the Reviewer), we now make real distinction between parameters such as organ size/volume, organ shape (based on organ outline), internal anatomy, and cellular content; all these terms (morphology, shape...) have been mixed and used in many different ways in previous studies because of the focus on single parameters (particularly size/volume). We agree that the internal anatomy of the brain has been published to some extent in an ecological context, as already highlighted in the second paragraph of the “Introduction”, but to our knowledge, very few quantitative analysis of brain shape (i.e., based on organ outline) has been published across vertebrate species, particularly using non-endocast material and squamate species. We now revised and clarified these aspects in the revised “Introduction” section, by also better highlighting the integrative approaches used in our study to assess brain evolution at various levels of biological organization.

The choice of cerebellum as a brain region to focus on is not sufficiently justified. There is no evidence that is “the most powerful model system for studying brain evolution”. In fact, there is still relatively little comparative research on cerebellar anatomy, despite the detailed descriptions of Larsell, who first revealed the neuroanatomical diversity of the cerebellum across vertebrates. Further, whether the cerebellum can be identified on an endocast or not is irrelevant to the study of cerebellar evolution across extant species.

We agree that there is no evidence that the cerebellum might be “the most powerful model system”, and we now revised this sentence to better justify the choice of the cerebellum based on both its key functions and great morphological diversity reported across vertebrates (including based on Larsell’s initial descriptions now cited in this paragraph). Similarly to the different approaches used on whole-brains (Histology, MRI, CT-scan etc...), the endocast-based methods are different but still complementary approaches to assess brain evolution in terms of size/volume and shape, including in extant species from all major vertebrate groups (see, e.g., Kawabe et al. *J. Anat.* 223:495, 2013; Ahrens *Anat. Rec.* 297:2318, 2014; Aristide et al. *Proc. Natl. Acad. Sci. U.S.A.* 113:2158, 2016; Marugán-Lobón et al. *J. Anat.* 229:191, 2016; Benson et al. *J. Anat.* 231:990, 2017; Allemand et al. *J. Anat.* 231:849, 2017; Watanabe et al. *J. Anat.* 234:291, 2019). We agree that volumetric and stereological analyses of animal brains have been more numerous and conclusive so far in the context of cerebellar structure, and we already removed most references to endocasts in the first revision of the manuscript, including in the revised “Abstract”, but we still consider important to shortly mention/highlight this work in our study (see also comment 3 below).

Several other statements were unclear or vague, such as “cellular content”, “multi-level, interspecific variations” and the entire sentence on page 3 (lines 10-14). I think what they are trying to say is that there is a lot interspecific variation, but the extent to which this is reflected in numbers or sizes of neurons, morphology, behaviour or ecology is unclear. This is not, however, how this comes across in this paragraph.

The statements and sentences mentioned by the Reviewer have now been revised.

In the last paragraph of the Introduction, I will first note that the cerebellum is not a “trait” in this context. The authors also do not explain what is meant by “multi-level specializations” and the entire sentence does not read like a hypothesis. This paragraph also lacks any predictions or a sense of what will be specifically tested in this study, including the methods to be used.

We now revised the last paragraph of the “Introduction” section according to this comment (and also based on comment 7 on transcriptomics), including the sentence explaining our hypothesis.

2. One of my main concerns with the analyses presented is the source of the brain material. First, no information is provided on fixation and different fixatives can have different effects on brain morphology and the amount of shrinkage relative to fresh brains. Second, museum collections typically fix specimens in formalin, but then store them long term in 70% ethanol. If the museum specimens were stored in ethanol, this can cause major changes in brain morphology and brain region sizes as a result of much greater dehydration than keeping brains in formalin. If this was the case here, the museum specimens could be biasing the analyses or providing inaccurate reconstructions of brain and cerebellar shape. Third, using a variety of different fixatives and storage procedures could affect the analyses across species, especially if the majority of a locomotor-group or clade came from a single source. The Supplementary Material lacks information on sources of the material, fixatives and other details, so it was not possible to evaluate if these potential confounds were affecting their

data. Fourth, using contrast enhancers can also introduce shrinkage and other artifacts that are not discussed and accounted for. If this is layered on top of the fixation/preservation issues I mention above, then there might be a range of artifacts affecting cerebellar morphology that could impact the results. These issues need to be addressed in order to have confidence in the results as presented.

We now provide the source of material in the revised Supplementary Table 1, and more information on fixation and staining procedures are given in the revised “Results and Discussion” and “Methods” sections. All specimens were fixed with formaldehyde-based solution, including 10% formalin for museum specimens and 4% paraformaldehyde for fresh specimens, which give almost identical final concentrations of formaldehyde. Museum samples were only conserved in formalin and not fixed and/or preserved in ethanol (we agree that very long-term ethanol immersion could result in a prolonged shrinkage of the tissue), and all contrast enhancer staining procedures were performed in the laboratory, thus limiting potential variations between samples highlighted by this reviewer. We agree that some shrinkage is of course possible during the fixation steps, but limited shrinkage of the brain is expected from formaldehyde-based solution at such low concentrations used (Fox *J. Histochem. Cytochem.* 33:845, 1985; Hughes et al. *PLoS one* 11:e0155824, 2016), and different samples per species have been used in our study whenever possible (as already mentioned in Methods). In fact, when available, we even directly compared our 3D models with freshly dissected brains, to confirm both our overall fixation/staining procedures and brain reconstructions (see the new Figure 2a); similarly, comparisons of 3D models between fresh and museum samples were also possible for some species, thus ensuring that the origins of the samples do not bias our data. The accuracy of our 3D reconstructions and landmarks across species was further validated by controlling for shape outliers in the geometric morphometric software MorphoJ (as now mentioned in Methods). Importantly, based on previous quantitative analysis of the brain using geometric morphometrics and histology, only minor shape and cytoarchitecture artifacts are expected using formaldehyde-based fixatives, including by comparing specimens with different preservation procedures (Weisbecker *Brain Struct. Funct.* 217:677, 2012; Hughes et al. *PLoS one* 11:e0155824, 2016). Museum samples have been used and published for many years in the vertebrate evolution field for both soft and bone tissues, and our guarantee on the fixation/conservation procedure is not expected to bias the validity of our data; especially, we don't expect our observed significant correlations with locomotor behavior to be only due to the presence of museum specimens, which are randomly distributed in our dataset and not linked to a particular ecology.

We also agree that contrast enhancers could potentially introduce some soft tissue shrinkage, even if these methods have been specifically optimized (including with optimized concentration and timing of staining, see below), refined, and used for visualizing soft tissue since more than 10 years, including in brain tissue, quantitative, and anatomical analyses. In fact, shrinkage has been shown to be either absent (including in museum specimens; Hedrick et al. *Microsc Microanal.* 24:284, 2018) or dependent on high concentrations of contrast agents (Vickerton et al. *J. Anat.* 223:185, 2013; Baverstock et al. *J. Anatom.* 223:46, 2013), explaining why only very low concentrations of contrast enhancers with optimized protocols and known penetration power (Metscher *Dev. Dyn.* 238:632, 2009; Metscher *BMC Physiol.* 9:11, 2009; Pauwels et al. *J. Microsc.* 250:21, 2013; Gignac et al. *J. Anat.* 228:889, 2016) were used in our study to specifically avoid or limit soft tissue shrinkage. Furthermore,

the rate and amount of shrinkage also varied with tissue type, probably due to differences of cellular organization and composition, altering the diffusion into and out of the tissue. Importantly, independent groups have observed that brain tissues such as the cerebellum undergo limited shrinkage or shape variation following contrast enhancer staining (Weisbecker *Brain Struct. Funct.* 217:677, 2012; Vickerton et al. *J. Anat.* 223:185, 2013), including in museum samples (Hedrick et al. *Microsc Microanal.* 24:284, 2018), likely as a result to the small cell size and/or high neuron density of these tissues. All these important justifications on the different procedures (including with new references, see references 46-55) are now further reinforced in the revised “Results and Discussion” and “Methods” sections of the manuscript.

3. Throughout the Results, there is a heavy emphasis on the Supplementary Material. Although I understand that journals such as this have strict page limits, the 20+ references to the Supplementary Material indicate that Supplementary Figures 1, 3 and 4 need to actually be embedded in the text of the manuscript proper.

Supplementary Figures 1, 3 and 4 are now embedded within the main manuscript (see new Figures 2, 3 and 4).

There are also some inaccuracies in the text of the Results. For example, the authors claims that they “quantified the accurate shape of the brain” for the first time in a squamate. This is not true; Hoops et al. (2018) published an MRI brain atlas of a lizard (<https://doi.org/10.1002/cne.24480>) that is far more detailed in its anatomical description than what is provided here. The reference to endocast analysis is also irrelevant here.

We agree to delete the “accurate” term in this sentence (and in others, see for example comment 8 below), because no comparisons with alternative methods such as MRI are presented in our study. However, we disagree with the rest of this comment: i) as also mentioned by the reviewer, the publication by Hoops et al. 2018 is a “detailed anatomical description” of a lizard brain which, although very useful and nicely illustrated, do not show any quantitative “shape” data (based on organ outline) or geometric morphometric analysis (we now give more precision on our “shape” parameter in the “Results and Discussion” section, see also reply to comment 1), ii) both whole-brain and endocast approaches have been important in the field to assess the ecological significance of brain size and morphology in multiple vertebrate groups, and a recent publication has demonstrated the overall strong size and shape correlations between most brain subdivisions and endocasts (Watanabe et al. *J. Anat.* 234:291, 2019). Similarly to the different methods used on whole brains (Histology, MRI, CT-scan etc...), the endocast-based methods are different but still complementary approaches to assess brain evolution, including for extant species from all major vertebrate groups (see reply to comment 1, including references).

I will add that the first part of the Results section was extremely difficult to read because it is presented as a single paragraph 3+ pages long with multiple reference to the Supplementary Material.

We agree with this comment and our initial brain description is now divided into two sections, one on overall brain morphology and one on quantitative shape analysis (with subparagraphs). Three figures from the Supplementary Material have also been embedded within the main manuscript (see above and new Figures 2, 3 and 4).

4. *One page 7, line 9 there is a reference to “data not shown”. This needs to be shown to demonstrate that this statistical assumption was violated and it could easily be included in the Supplementary Material. On the same page, line 23 there is a reference to “other parameters”. This needs to be spelled out as there are some variables examined in this manuscript and many others that are not (e.g., neuron numbers, neuron sizes, neuronal complexity, neuropil volume).*

The statistical test on homogeneity of regression slopes (corrected for multiple comparisons based on Bonferroni method) is now shown in the new Supplementary Table 6. “Other parameters” was referring to the shape in this case (as evident from next paragraph), and this aspect is now clearly mentioned in the revised text. Other interesting neuronal aspects mentioned by the reviewers (also based on comment 8) are now included in our revised “Results and Discussion” and “Conclusions” sections as future project directions.

5. *In the analysis of Purkinje cell layout, some of the findings were over-stated. For example, the “remarkable” variation in Purkinje cell layer organization across squamates has been described before. The authors acknowledge that part-way through this section, but the description of this variation needs to be tempered to reflect the previous observations. I was also confused by what was means by “directly comparing the obtained four groups with similar statistical analysis”. Last, the link between locomotor disorders and squamate locomotion seems a bit of a stretch. Yes, some locomotor disorders are associated with ectopic Purkinje cells, but this makes it sound as if the evolution of burrowing/lateral undulating is the result of cerebellar disorganization when that is unlikely the case. The evolution of different locomotor strategies might require different kinds of cerebellar mediated coordination, resulting in a different topological organization of the Purkinje cells, which is different to abnormal cerebellar development in mutant mice or humans with neurological disorders.*

We only partially agree with this Purkinje cell (PC) comment, as although the variability in PC spatial organization has been described to some extent in some species, only a few squamate lizard and snake species showing “extreme” phenotypes (i.e., ordered monolayer versus totally scattered) were tested in these studies. Our new analysis across a wide range of species now reveals “intermediate” spatial configurations within squamates as well as different degrees of scattering within both snakes and lizards. The “Results and Discussion” section on Purkinje cells has now been edited to better reflect all these facts; for example, based on the comment on previous reports, we now first describe previous observations made on Purkinje cell arrangement, and we clearly mention that our observations “confirm” these preliminary data. The sentence mentioning “similar statistical analysis” is better defined in the same paragraph as well as in the revised “Methods” section. Finally, we revised the correlations between mouse/human locomotor disorders and squamate locomotion in our discussion based on Reviewer’s comment.

6. *On page 9, lines 4-7, I could not determine what the authors were trying to say here and “personal observations” are not sufficient for a major journal article to support this statement.*

We fully agree that this part does not help in justifying our transcriptomic analysis, which is in fact more related to the next sentence on the transcriptome-phenotype relationships. Furthermore, the introduction part for this transcriptome paragraph is relatively long, so the whole sentence was deleted to avoid any mis-interpretation.

7. *Although the gene expression part of this paper is better described in this version, lines 9-12 on page 9 are better suited to the Introduction, which lacks any description or mention of transcriptomics. I*

would also say again, that linking some of the findings to mutant mice is a stretch and implies here that limbless species are uncoordinated, when they are not. Limbless species have a different type of coordination than quadrupedal squamates because the locomotor pattern is different.

As suggested, the sentence on transcriptomics mentioned by the Reviewer is now in the revised “Introduction” section, as a similar sentence is in fact mentioned later in the same paragraph when introducing the hierarchical clustering method. The addition of gene examples identified in our analysis is based on the comments of Reviewer 2, in order to better confirm the validity of our transcriptome analysis (including with justification of functional importance based on example of knock-out animal models). Our data only indicate that genes previously linked to motor control, balance, and/or locomotor activity exhibit different expression levels across squamates, and we never show or imply that our squamate samples have deleted forms of these genes (in contrast to our previous discussion regarding the PC phenotype which might have given this impression; this part is now revised, see reply to comment 5). In fact, and related to the comment 5 of this reviewer, our data rather suggest that the evolution of different locomotor strategies might require expression level modulation of the identified cerebellar genes, indicating that differential gene expression might reflect the specific locomotor pattern in squamates. We now clarified this aspect in the “Results and Discussion” paragraph on gene expression data.

8. Like the Introduction, the Conclusions lacks focus and contains inaccurate and/or vague phrases. For example, there is no means of determining whether the current analyses are more or less accurate than any others. Further, the analyses presented are not as comprehensive as claimed. Currently, most studies of this nature quantify neuron numbers and there is a profound lack of information on variation in neuron numbers and neuronal morphology across species, for almost every vertebrate clade. This is not acknowledged anywhere in the manuscript, but should be discussed here and elsewhere. I will also add that the current study does not actually reflect “cerebrotype” analyses, which would be dependent on analyses across multiple brain regions or, at the very least, quantification within the cerebellum.

As suggested by the reviewer, a more nuanced and precise discussion of our findings and conclusions is now presented. Terms such as “accurate”, “considerable”, “substantial”, and “multiple” have been either removed or replaced. We agree that additional levels of cerebellar complexity could be also assessed, and we now introduced the new interesting neuronal aspects mentioned by this Reviewer (based also on comment 6) in the “Results and Discussion” and “Conclusions” sections as future project directions. As seen in new Figure 4 (but also in Supplementary tables 2 and 3), our comparative shape analysis based on geometric morphometrics has in fact been done “across multiple brain regions”; so, based on the comment of this Reviewer, we would consider this part of the work as a “cerebrotype analysis”. However, we still clarified the “cerebrotype” aspect in our revised “Abstract” and “Conclusions” sections, as we agree that the term “cerebrotype” has been used for size/volume data in previous analyses, and our analysis involving each individual brain subdivisions rather show organ shape data. However, based on the definition “brain pattern shared by groups of species with lifestyle similarities”, we are still convinced that our new shape data demonstrate and even amplify the concept of “cerebrotype”. Similarly, we further clarified the mosaic aspect, which is only based on shape data presented across multiple brain regions.

9. The methods describes manual segmentation about several brain regions, but there is no actual analyses of these presented. Morphometrics can be done without segmenting individual brain regions, so an explanation is needed for segmenting non-cerebellar brain regions and an analysis provided (which would be a cerebrotype-analysis), or some of this methods section removed.

This comment is not clear to us, as also related to the point 8 above, geometric morphometrics has been in fact done for the whole-brain but also for each individual brain subdivisions (see new Figure 4 as well as Supplementary tables 2 and 3). If the criticism is about the lack of “analyses across multiple brain regions” (as mentioned above in point 8), this is not correct as done in our study at least for the shape (see new Figure 4). If the criticism is about the segmentation method used, we now clarified this issue in the revised “Methods” of the manuscript as geometric morphometrics has in fact been done on each brain subdivisions using different specific subsets of landmarks on the whole-brain reconstructions (as also evident from both new Supplementary Figure 1 and Methods section on geometric morphometrics). Based on this comment, we also clarified the “cerebrotypes” aspect in our revised “Abstract” and “Conclusions” sections (see also reply to comment 8).

10. Minor issues.

- a. There were at least two instances of “However, although”, which is grammatically incorrect.*
- b. “neuron organization” is not a term*
- c. Page 4, lines 29-30. The sentence that begins on line 29 needs to be rephrased.*

These minor issues have been corrected.

Reviewers' Comments:

Reviewer #3:

Remarks to the Author:

I am pleased that my primary concern, the fixation of tissue samples, was not a significant factor. Also, the additional information on methods is greatly appreciated as this resolves a number of outstanding issues. Although I disagree with the authors on several of their statements concerning morphometric approaches in evolutionary neuroscience, I have no major concerns or criticisms of the revised manuscript.

I have a few minor corrections (page and line numbers refer to the marked version of the manuscript).

page 2, lines 31-34. This was a bit dense, so I would suggest breaking it up into a couple of sentences.

page 4, line 14. I am not sure that "read-out" is the most appropriate term to use here.

page 9, line 5. Replace "have already shown to be" with "is known to be"

Nature Communications manuscript NCOMMS-19-15106B

Multi-level variation in squamate cerebellum architecture is associated with locomotor specialization.

Simone Macrì, Yoland Savriama, Imran Khan & Nicolas Di-Poi

Point-by-point replies to the reviewers' and editorial comments and suggestions

We thank again the Reviewers for the positive evaluation of our manuscript. We have now incorporated the new minor corrections from Reviewer 3. We reproduce the referee's comments in italics and our responses and explanations are in plain text.

Reviewer #3 (Remarks to the Author):

I am pleased that my primary concern, the fixation of tissue samples, was not a significant factor. Also, the additional information on methods is greatly appreciated as this resolves a number of outstanding issues. Although I disagree with the authors on several of their statements concerning morphometric approaches in evolutionary neuroscience, I have no major concerns or criticisms of the revised manuscript.

I have a few minor corrections (page and line numbers refer to the marked version of the manuscript). page 2, lines 31-34. This was a bit dense, so I would suggest breaking it up into a couple of sentences.

This long sentence has been now divided into three shorter sentences.

page 4, line 14. I am not sure that "read-out" is the most appropriate term to use here.

The term "read-out" has been replaced by "insights".

page 9, line 5. Replace "have already shown to be" with "is known to be"

This minor issue has been corrected.